# Enhancing polyol/sugar cascade oxidation to formic acid with defect rich MnO$_2$ catalysts

Hao Yan [1,2,8], Bowen Liu [3,8], Xin Zhou [1,4,8], Fanyu Meng[1], Mingyue Zhao[1], Yue Pan[1], Jie Li[1], Yining Wu[5], Hui Zhao[1], Yibin Liu [1]✉, Xiaobo Chen[1], Lina Li[6], Xiang Feng [1]✉, De Chen[7], Honghong Shan[1], Chaohe Yang[1] & Ning Yan [2]✉

Oxidation of renewable polyol/sugar into formic acid using molecular O$_2$ over heterogeneous catalysts is still challenging due to the insufficient activation of both O$_2$ and organic substrates on coordination-saturated metal oxides. In this study, we develop a defective MnO$_2$ catalyst through a coordination number reduction strategy to enhance the aerobic oxidation of various polyols/sugars to formic acid. Compared to common MnO$_2$, the tri-coordinated Mn in the defective MnO$_2$ catalyst displays the electronic reconstruction of surface oxygen charge state and rich surface oxygen vacancies. These oxygen vacancies create more Mn$^{\delta+}$ Lewis acid site together with nearby oxygen as Lewis base sites. This combined structure behaves much like Frustrated Lewis pairs, serving to facilitate the activation of O$_2$, as well as C–C and C–H bonds. As a result, the defective MnO$_2$ catalyst shows high catalytic activity (turnover frequency: 113.5 h$^{-1}$) and formic acid yield (>80%) comparable to noble metal catalysts for glycerol oxidation. The catalytic system is further extended to the oxidation of other polyols/sugars to formic acid with excellent catalytic performance.

Formic acid (FA), as the simplest carboxylic acid, is a valuable chemical and a potential energy carrier[1,2]. Preferably, FA is obtained from renewable resources such as biomass or CO$_2$ rather than the current route based on fossil derived feedstock (e.g., methanol and CO)[3–5]. In this context, catalytic oxidation of polyol/sugar emerges as a promising alternative for FA production[6–16]. However, the mainstream homogeneous catalysts including heteropoly acids, NaVO$_3$, VOSO$_4$, and others suffer from poor catalyst recycling, metal contamination, and/or the employment of non-benign oxidants[8,17–23]. Recently, several heterogeneous catalysts based on CuO$_x$ or VO$_x$ have been excavated using O$_2$ as oxidant[20,24], but the attainment of a high FA yield with sufficient reaction rate using non-precious metal oxide catalyst

remains a challenge[1]. One limitation is the sluggish O$_2$ activation and subsequent bond cleavage over the saturated metal oxide surface with few accessible sites[4,19]. Developing defective metal oxide catalysts with enhanced ability for bond activation is desirable in the cascade oxidation of polyol/sugar to FA using molecular oxygen.

Manganese oxides exhibit a wide range of applications in oxidation reactions due to the superior redox property and structural flexibility[25,26]. Mn-based oxides, mainly MnO$_2$, are amphoteric, containing Mn-induced acidity and nearby O-induced basicity. These co-existing acidic and basic sites in metal oxides could be regarded as frustrated Lewis pairs (FLPs), which promote reactions on the catalyst surfaces that require both acid and base functions, as previously

[1]State Key Laboratory of Heavy Oil Processing, China University of Petroleum (East China), Qingdao 266580, China. [2]Department of Chemical and Biomolecular Engineering, National University of Singapore, Singapore Engineering Drive 4, 117585, Singapore. [3]Department of Chemistry, University of Liverpool, Crown Street, L69 7ZD Liverpool, UK. [4]College of Chemistry and Chemical Engineering, Ocean University of China, Qingdao, Shandong 266100, China. [5]School of Petroleum Engineering, China University of Petroleum (East China), Qingdao 266580, China. [6]Shanghai Synchrotron Radiation Facility, Shanghai Advanced Research Institute, Chinese Academy of Sciences, Shanghai 201204, China. [7]Department of Chemical Engineering, Norwegian University of Science and Technology, Trondheim 7491, Norway. [8]These authors contributed equally: Hao Yan, Bowen Liu, Xin Zhou. ✉e-mail: liuyibin@upc.edu.cn; xiangfeng@upc.edu.cn; ning.yan@nus.edu.sg

demonstrated on (hydroxylate-)$Al_2O_3$, $InO_x$, $CeO_x$, $MoO_x$ and $CoO_x$ oxides[27–35]. We notice that the cascade oxidation of alcohol using $O_2$ may be regarded an acid-base synergistic reaction: the activation of $O_2$ into reactive oxygen species is accelerated by acid sites[36,37], while C–H bond activation and C–C bond cleavage are known to be promoted by basic sites[38,39]. The characteristics of FLPs match the active site requirements for polyol/sugar oxidation[40–44], but the saturated coordination structure in traditional $MnO_2$ catalyst shows limited surface oxygen vacancies and low surface electron density. Due to the tendency of the rigid lattice of traditional metal oxides to form saturated coordination M–O structures, it is still challenging to construct defective metal oxides with FLPs based on existing limited and cumbersome preparation strategies[45–47]. Inspired by the variable coordination structure of transition metals[48], rational reduction of the coordination number of the central metal provides a feasible strategy to obtain defective $MnO_2$ with enhanced FLPs for polyol/sugar cascade oxidation to formic acid.

Herein, we present a strategy to construct low-coordinated, defective $MnO_2$ catalyst ($MnO_2$-D) for polyol/sugar oxidation. Specifically, the defective $Mn^{\delta+}$-$O_V$ structure with unsaturated tri-coordinated Mn forms FLPs to allow spatially adjacent acid and basic sites to work cooperatively. $Mn^{\delta+}$ species associated with oxygen vacancy serve as Lewis acid to promote the activation of $O_2$ under the assistance of adjacent basic sites, whereas electron reconstructed O nearby $Mn^{\delta+}$ serves as Lewis base to facilitate the bond activation of organic substrate, in synergy with the Lewis acid site. In this manner, $MnO_2$-D with intensified FLPs exhibits superior catalytic activity in converting various substrates into formic acid using $O_2$, surpassing previous reports using heterogeneous catalysts.

## Results

### Synthesis and characterization of defective $MnO_2$-D with low Mn-O coordination

Two α-$MnO_2$ catalysts ($MnO_2$-P and $MnO_2$-D) were synthesized by hydrothermal methods, in which P and D refer to perfect and defective structures, respectively[49,50]. Significant differences in composition and crystal structure were observed on Rietveld refinement of powder X-ray diffraction (PXRD) and high-resolution transmission electron microscopy (HRTEM) images. Although only the Bragg peaks corresponding to the α-phase $MnO_2$ (I4/m space group) are present for both samples (Fig. 1a)[51–54], the PXRD pattern of $MnO_2$-D displays a broader diffraction peak than that of $MnO_2$-P, indicating its smaller crystallite size [8.2(1) nm] compared with $MnO_2$-P [36.5(1) nm]. The site occupancies of O and Mn were obtained by refining the PXRD patterns (Supplementary Tables 1 and 2), which were then taken as the starting model for Density functional theory (DFT) calculation described in later sections. The hydrothermal synthesis process was significantly affected by the nature of the reductant and temperature, resulting in a sharp difference in the amount of O and Mn ion defects in the two catalysts. In $MnO_2$-P, Mn is distributed at the octahedral 8 h site with a 99(1)% occupancy, along with the existence of oxygen vacancy [5(2)%] at $O_1$ (8 h) site. The valence state sum of Mn cation was calculated as 3.93+ based on the charging balance. In contrast, in $MnO_2$-D the Mn occupancy of Mn at the octahedral 8 h site is 99(2)% with 13(1)% oxygen vacancy at $O_1$ (8 h) site and 7(2)% oxygen vacancy at $O_2$ (8 h) site, with the valence state of Mn determined at a lower value of 3.64+. The formation of more defect structures in $MnO_2$-D also resulted in the change of Mn–O chemical bond length (Supplementary Fig. 1), which manifests the Jahn–Teller (JT) effect[55–57]. The Mn–O bond lengths in $MnO_2$-P are Mn–$O_1$: 1.64(3) Å and Mn–$O_2$: 1.89(3) Å, respectively, while the ones in $MnO_2$-D are Mn–$O_1$: 1.69(4) Å and Mn–$O_2$: 1.92(4) Å[51,58,59]. Scanning electron microscopy (SEM) image in Fig. 1b shows that $MnO_2$-P consists of short and thick nanowires with length of ~500 nm and width of ~30 nm, while $MnO_2$-D exhibits longer and thinner nanowire morphology (width: ~10 nm), which are consistent with the crystallinity

results of PXRD, and the notion that $MnO_2$-D exposes more defective sites on surface. The HRTEM images and selected area electron diffraction (SAED) pattern indicate that $MnO_2$-P mainly exposes (211) and (200) planes, while $MnO_2$-D mainly exposes (310) and (301) planes (Fig. 1b). The crystallinity results of PXRD also confirms that the ratio of (310) crystal plane to (211) crystal plane on $MnO_2$-D [$I_{(310)}/I_{(211)} = 0.78$] is significantly higher than that on $MnO_2$-P [$I_{(310)}/I_{(211)} = 0.65$].

To confirm the coordination structure and valence states of Mn, X-ray absorption fine structure (XAFS) were performed. The $k^3$-weighted Fourier-transform Mn K-edge extended XAFS spectra (EXAFS) in Fig. 1c show that both $MnO_2$-P and $MnO_2$-D exhibit typical spectral features of α-$MnO_2$ phase with the intense FT peaks at approximately 1.52 Å, 2.85 Å and 3.40 Å, corresponding to Mn–O, edge-shared Mn–(O)–Mn, and corner-shared Mn–Mn shells, respectively[55,56,60,61]. Compared with $MnO_2$-P, $MnO_2$-D exhibits lower intensity of Mn-O shell and stronger intensity of corner-shared Mn-Mn shell, manifesting the increase of defective structure[49,50,62]. Wavelet transform (WT) analysis in Supplementary Fig. 2 shows that the peak with a maximum intensity of approximately 6 Å is observed for bulk α-$MnO_2$, $MnO_2$-P and $MnO_2$-D in the k-space, ascribed to the Mn–O bond. The WT spectrum of $MnO_2$-D also shows a maximum intensity of approximately 12 Å in the k-space and 3.5 Å in the R-space, corresponding to the Mn-Mn shell. EXAFS data fitting suggests that the coordination number of O for bulk α-$MnO_2$ and $MnO_2$-P is 6.0 and 5.2, respectively, while that for $MnO_2$-D is only 3.0, confirming the significant decrease of Mn-O coordination (Supplementary Table 3). Figure 1d provides a cartoon representation of the 5-coordination pentahedron geometry of Mn in $MnO_2$-P, and the 3-coordination plane/tetrahedron structure of Mn in $MnO_2$-D. The latter is an equivalent model consisting of saturated 6-coordination $Mn^{4+}$ and other defect coordination $Mn^{\delta+}$. Obviously, compared with penta-coordinated Mn in common $MnO_2$-D, the tri-coordinated Mn is successfully constructed in the defective $MnO_2$-D.

Following exhaustive characterizations of $MnO_2$-D and $MnO_2$-P, we proceeded to synthesize tetra-coordinated α-$MnO_2$ ($MnO_2$-T) with an intermediate coordination number of 4.2 (Supplementary Fig. 2 and Supplementary Table 3). In addition, we synthesized and characterized two sets of distinct phases, β-$MnO_2$ and γ-$MnO_2$. Each set comprises a high coordination and a low coordination sample, respectively (Supplementary Table 3 and Supplementary Fig. 2). The intention behind these preparations was to establish a comprehensive comparison on glycerol oxidation efficiency across various $MnO_2$ samples bearing different coordination environment.

### Probing the structure of the Frustrated Lewis Pairs in defective $MnO_2$-D

The Mn K-edge X-ray absorption near edge structure (XANES) shows that the absorption edges of $MnO_2$-P and $MnO_2$-D are similar to standard $MnO_2$ (Fig. 2a). Through the first-derivative of absorption edge in normalized XANES spectra, the Mn valence state is quantified as 3.89 and 3.56 for $MnO_2$-P and $MnO_2$-D respectively (Supplementary Fig. 3), consistent with PXRD refinement results. The Mn 2p X-ray Photoelectron Spectroscopy (XPS) spectra (Fig. 2b) clearly demonstrates that the formation of $Mn^{\delta+}$-$O_V$ defect structure leads to a significant decrease of the binding energy, confirming that the surface of $MnO_2$-D possesses more electrons[63]. The XPS peak deconvolution analysis in Supplementary Table 4 shows a higher content of $Mn^{2+}$ and $Mn^{3+}$ is observed in $MnO_2$-D. This electronic reconstruction caused by the reduction of coordination number induces Mn to be more prone to get electrons, while O more prone to lose electrons—an essential prerequisite for the formation of FLPs.

Notably, the generation of oxygen vacancy in the low-coordination $Mn^{\delta+}$-$O_V$ structure is crucial for FLPs formation: the reduced oxygen coordination leads to the exposure of the metal atom as Lewis acid sites, while the oxygen vacancies also induce

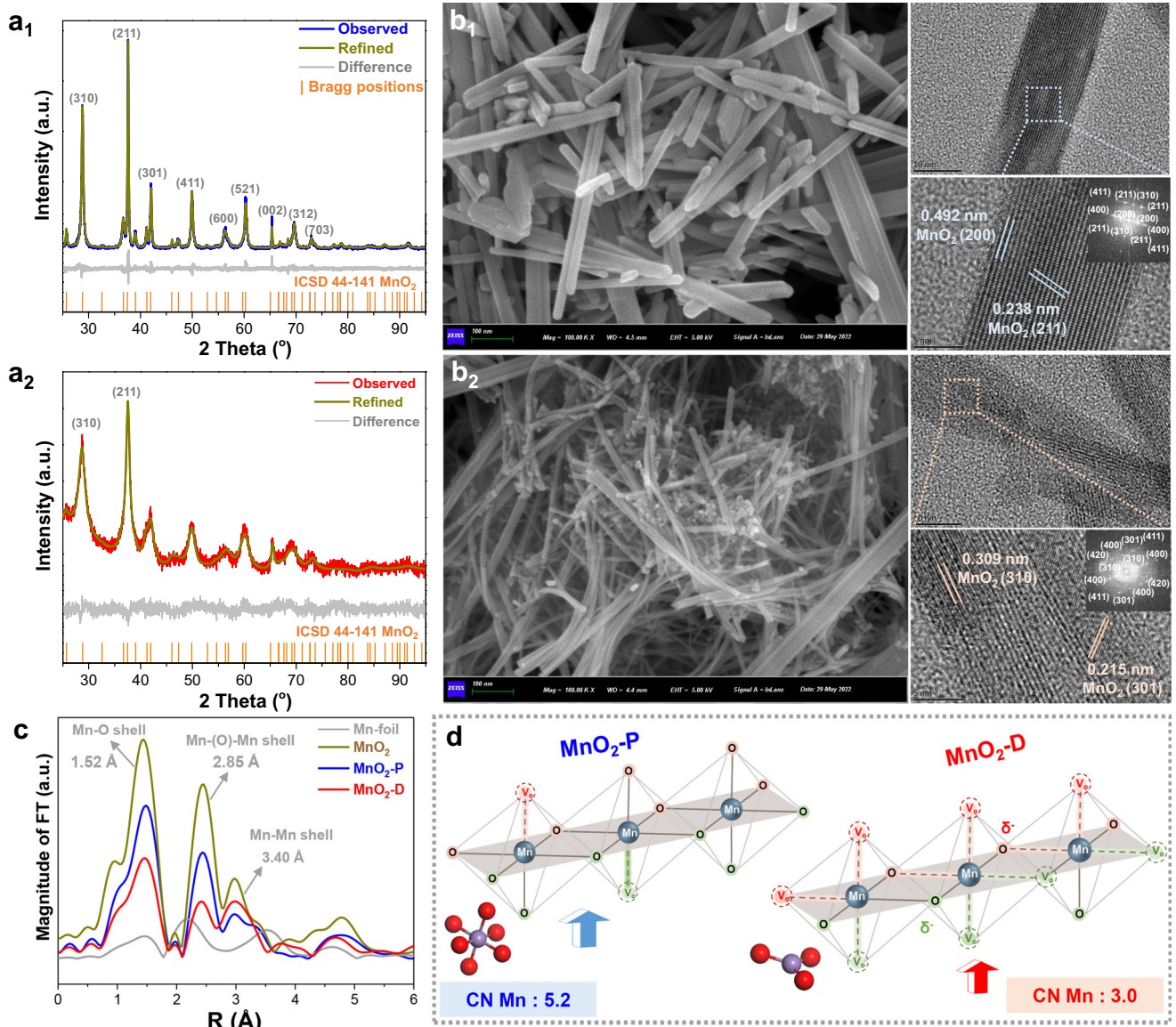

**Fig. 1 | Coordination structure of MnO₂-P and MnO₂-D. a** Rietveld refinement and visualization of the associated crystal structure for **a₁** MnO₂-P and **a₂** MnO₂-D. Unit cell of samples MnO₂-P (I4/m (84), a = b = 9.804(1) Å, c = 2.855(1) Å) and MnO₂-D (I4/m (84), a = b = 9.818(1) Å, c = 2.847(1) Å) obtained from powder diffraction data based on the Rietveld profile refinement and Stephens peak shape function using the Topas software (Supplementary Tables 1 and 2) with the $R_{wp}$ are 11.62% and 9.82%, respectively. **b** SEM/HRTEM images of **b₁** MnO₂-P and **b₂** MnO₂-D. **c** Fourier transform of Mn K-edge extended EXAFS oscillations. **d** Schematic diagram of coordination structure of Mn$^{\delta+}$ on MnO₂-P and MnO₂-D.

stronger electronegativity of the residual oxygen, thus enhancing the Lewis basicity. As shown in Fig. 2c, the temperature-programmed desorption (TPD) of O₂ of the two catalysts reveals the presence of the surface oxygen species (~200 °C, $O_{sur}$), the active oxygen species near the surface (~300 °C, $O_{sur-sub}$) and the lattice oxygen species (~600 °C, $O_{lat}$). MnO₂-D exhibits a high $O_{sur-sub}$ content (77.6%) and a minimum $O_{lat}$ content (9.5%), indicating that the defective structure induces more active oxygen species (Supplementary Table 5). Low-temperature electron paramagnetic resonance (EPR) also shows that MnO₂-D exhibits a stronger oxygen vacancy feature at a g-factor of 2.003[64,65] than MnO₂-P (Supplementary Fig. 4). Consistent with O₂-TPD and EPR analysis, the XPS spectra of O 1s region in Fig. 2d and Supplementary Table 4 demonstrates that the MnO₂-D catalyst has the highest oxygen vacancy content (27.6% $O_{II}$) with lower binding energy. Additionally, the existence of oxygen vacancy on the surface of MnO₂-D is observed via aberration-corrected high-angle annular dark-field scanning TEM (Fig. 2e).

On these foundations, we analyzed the structure and Mulliken charge of penta-coordinated MnO₂-P and tri-coordinated MnO₂-D to provide direct evidence for the formation of FLPs. Figure 2f shows that the near saturated coordination O in MnO₂-P creates a steric hindrance effect that obstructs the access of partially negatively charged reactant molecules to the Mn Lewis acid site. In sharp contrast, Mn in the low-coordinated MnO₂-D being highly unsaturated requires more molecules to coordinate (Lewis acid property). In parallel, the three O atoms in MnO₂-D process more electrons due to the electron transfer from Mn to O, which could further transfer electrons to reactant molecule (Lewis base property). Hence, the behavior of defective Mn$^{\delta+}$–$O_V$ structure in MnO₂-D aligns well with the concept of FLPs[30,66–70]. To probe the FLPs on MnO₂-D, NH₃-TPD, and CO₂-TPD were further performed. Figure 2g and Supplementary Table 5 show that the MnO₂-D exhibits sharply different acid-base properties from MnO₂-P. For NH₃-TPD, MnO₂-D displays more medium strong and strong acid sites, while MnO₂-P mainly contains weak acid sites with a small amount of strong acid sites. This indicates that the decrease of coordination

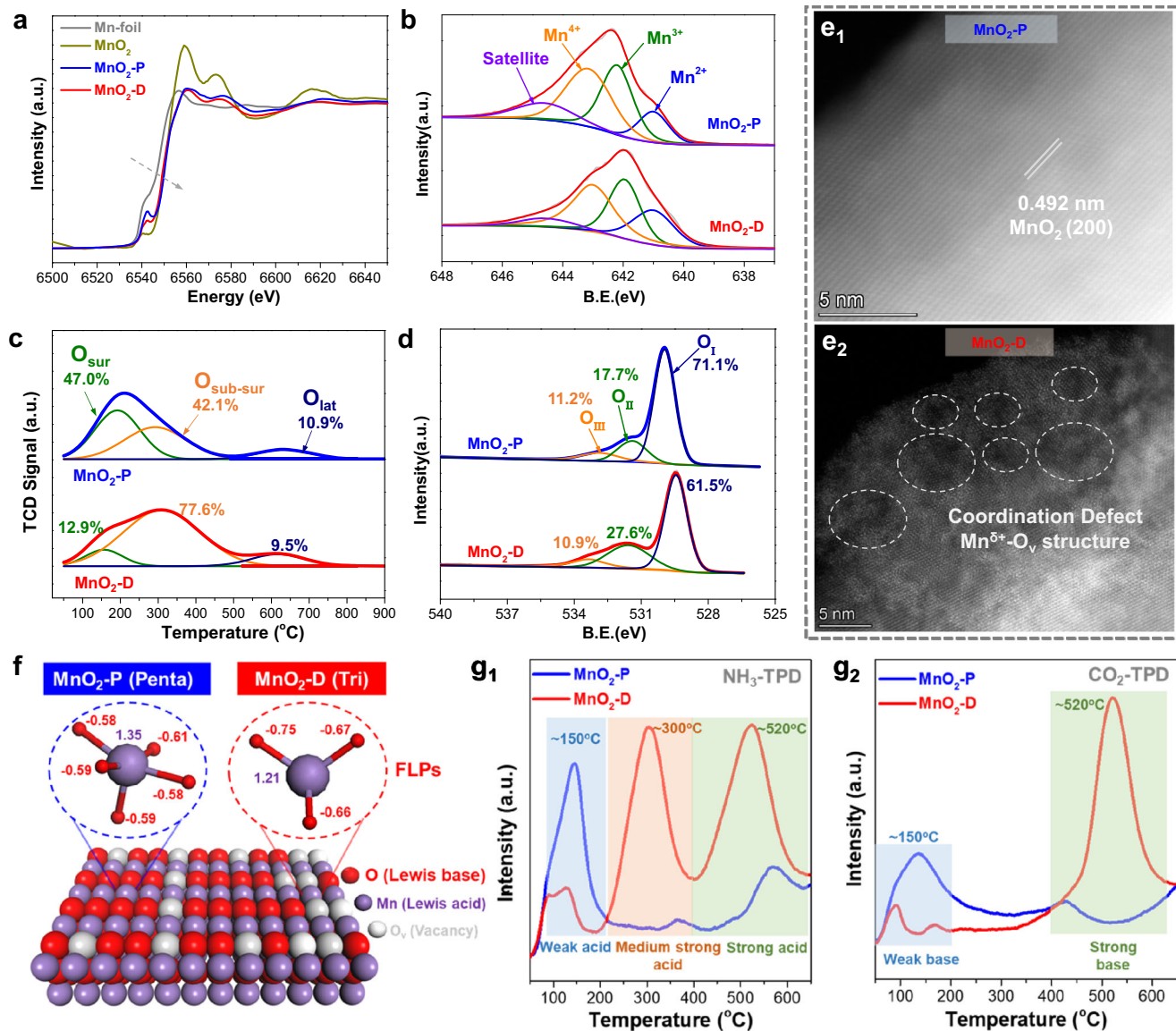

**Fig. 2 | Strength-intensified Frustrated Lewis Pairs of MnO₂-D. a** The first-order derivatives of Mn K-edge XANES. **b** Mn 2p XPS spectra of MnO₂-P and MnO₂-D. **c** O₂-TPD and **d** XPS spectra of O 1s of MnO₂-P and MnO₂-D. **e** Aberration-corrected HAADF-STEM image of **e₁** MnO₂-P and **e₂** MnO₂-D. **f** Structure diagram and Mulliken charge distribution of Frustrated Lewis Pair in MnO₂-P and MnO₂-D. **g** NH₃-TPD and CO₂-TPD of **g₁** MnO₂-P and **g₂** MnO₂-D.

number not only increases the amount of Lewis acid sites (exposure of more Mn), but also increases the acid strength. In addition, pyridine infrared results demonstrate that the Lewis acid sites on MnO₂-P and MnO₂-D are water tolerant (Supplementary Fig. 5). In CO₂-TPD, MnO₂-D exhibits increased amount of strong base sites compared with MnO₂-P, attributable to the stronger adsorption of CO₂ on electron-enriched O in FLPs.

### Enhanced cascade oxidation of polyol/sugar to formic acid over MnO₂-D

Next, catalytic oxidation of glycerol was tested using the prepared Mn catalysts under 1 MPa O₂ at 120 °C. Figure 3a and Supplementary Table 6 show that at low conversion (<20%), the initial reaction rate reached 31.3 mmol/h/g$_{cat}$, almost 20 times higher than that of MnO₂-P. Moreover, at a similar glycerol conversion level (~64%), MnO₂-D exhibits a high FA selectivity of 83.3%, while MnO₂-P displays a total selectivity of 66.9% for various carboxylic acids, including glyceric acid (GLYA), oxalic acid (OA), glycolic acid (GLYOA) and FA. The tetra-coordinated MnO₂-T, characterized by its intermediate coordination

number, exhibited moderate activity that fell between the performances of MnO₂-P and MnO₂-D catalysts (20.7 mmol/h/g$_{cat}$). Consistently, low-coordination number β-MnO₂ and γ-MnO₂ samples displayed superior catalytic performance in oxidizing glycerol to FA compared to their high coordination number counterparts (Supplementary Table 6). The above results indicates that the saturated coordination structure of MnO₂ does not have sufficient activity in C–H and C–C bond activation, implying that an unsaturated structure is key for high activity. Of note, the difference of specific surface area is not the main factor affecting the activity and FA selectivity, as excluded by control experiments (Supplementary Fig. 6), while little activity was observed in the absence of a Mn-based catalyst (Supplementary Table 6). Through further optimization of reaction time, MnO₂-D delivered an optimal 99.2% conversion and 83.2% FA selectivity in 6 h (Supplementary Fig. 7).

Ethanol pulse adsorption was further conducted to determine the exposed active sites (Supplementary Fig. 8), based on which the TOF of MnO₂-D catalyst was determined as 113.5 h$^{-1}$. This value matches or outperforms the catalytic activities of noble metal catalyst operated

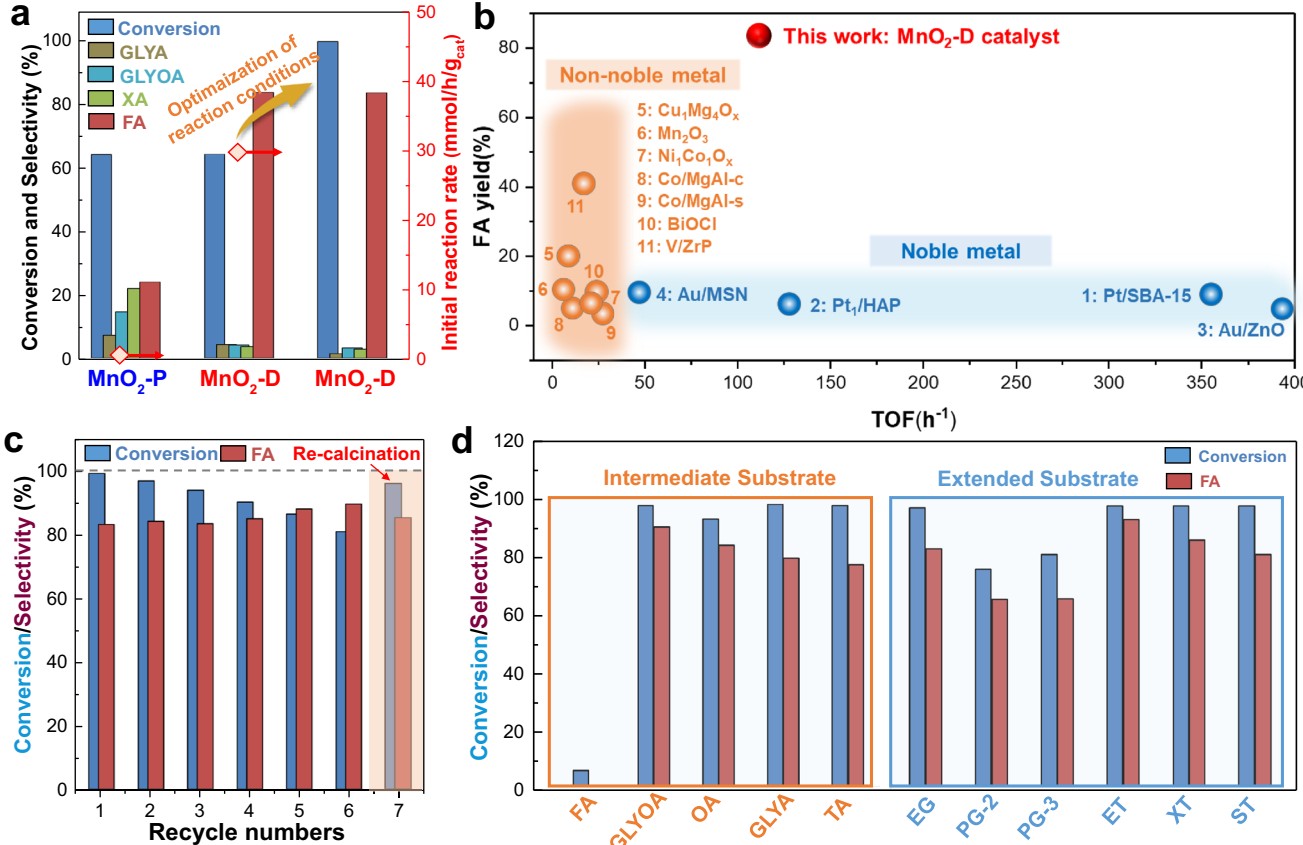

**Fig. 3 | Cascade oxidation of polyol/sugar to formic acid over MnO₂-P and MnO₂-D. a** Cascade oxidation of glycerol to formic acid over the MnO₂-P and MnO₂-D [25 mL aqueous phase solution (0.1 M), 0.5 g NaOH, 0.1 g catalyst, 1 MPa O₂, MnO₂-P for 10 h, MnO₂-D first for 2 h, MnO₂-D second for 6 h,120°C]. **b** Oxidation of glycerol to formic acid over MnO₂-D and other reported catalysts. **c** Catalytic stability of the MnO₂-P and MnO₂-D under multiple cycle test conditions. **d** Catalytic performance of the MnO₂-D for catalytic oxidation of different substrates (polyols/sugars) to formic acid [Formic acid (FA); Glycolic acid (GLYOA); Oxalic acid (OA); Glycerol (GLYA); Tartronic acid (TA); Ethylene glycol (EG); 1,2-propanediol (PG-2); 1,3-propanediol (PG-3); Erythritol (ET); Xylitol (XT); Sorbitol (ST)].

under similar conditions (Fig. 3b and Supplementary Table 7). Moreover, MnO₂-D exhibits excellent stability for glycerol oxidation, with almost no change in FA selectivity (>80%) and a slight decrease in glycerol conversion (<20%) after 6 catalytic cycles due to the coverage of some surface active sites by carboxylic acid products (Fig. 3c and Supplementary Fig. 5). Characterizations of the spent catalyst using XRD, UV-Vis, XPS, NH₃-TPD and CO₂-TPD techniques prove that the defective Mn$^{\delta+}$–O$_V$ structure and acid-base properties in MnO₂-D were well-preserved during the reaction (Supplementary Fig. 5). After calcining the spent catalyst, the catalytic activity of MnO₂-D was fully restored. MnO₂-D is further extended to the selective oxidation of intermediate substrates derived from glycerol and other polyols/sugars (Fig. 3d). In all cases, high conversion (>75%) and FA selectivity (>60%) were obtained. The excellent catalytic performance encourages us to further explore the structure-activity relationship on the defective Mn$^{\delta+}$–O$_V$ structure with strength-intensified FLPs in the polyol/sugar oxidation to FA.

## Activation of O₂ into hydroxyl radical on Lewis acid site in MnO₂-D

In situ EPR spectra were collected to probe the evolution of oxygen vacancy of MnO₂-D during catalytic oxidation of glycerol (Fig. 4a). The signal of oxygen vacancy could be regarded as the function strength of FLPs in the Mn$^{\delta+}$-O$_V$ structure since the strength-intensified FLPs are originated from the rich oxygen vacancies induced by the low-coordination structure. We observe a symmetrical EPR peak at g = 2.003 in MnO₂-D, attributable to unpaired electrons associated

with oxygen vacancies of metal oxides. Interestingly, the intensity of the peak shows a progressive increase with rising reaction temperature. Moreover, the peak intensity demonstrates a strong linear correlation with initial catalytic activity (Supplementary Fig. 9), indicating that an increase in the oxygen vacancy signal corresponds to a proportional increase in catalytic activity. The sluggish O₂ activation is one of the important reasons that restrict the oxidation activity. Inferentially, the formation of strength-intensified FLPs promotes the activation of O₂, and then linearly increases the oxidation activity. To confirm this point, free radical trapping agent was added during glycerol oxidation by MnO₂-P and MnO₂-D catalyst to provide insights on O₂ activation (Fig. 4b). Blank test confirmed that there is no signal of radical in the absence of catalyst. In sharp contrast, both catalysts (especially MnO₂-D) display obvious hydroxyl radical signals. The hydroxyl radical quenching experiment, employing organic and inorganic radical scavengers, further established that the produced hydroxyl radical significantly impacts the oxidation reaction (Supplementary Table 6). It is plausible that O₂ mainly interacts with H₂O to generate hydroxyl radicals that participate in the subsequent glycerol activation, aligning with previous studies[71–73].

Due to the difference in electronegativity between Mn$^{\delta+}$ and O$_V$, the d-π feedback of the anti-bonding orbital weakens the O = O bond, allowing O₂ to dissociate into OH* in the presence of H₂O (Fig. 4c). The activation of O₂ is mainly promoted by Mn$^{\delta+}$ acid site with the assistant of O$_V$ basic sites since one O atom is directly anchored by the Mn center, while another O atom is activated by H dissociated from H₂O. For H₂O activation, H₂O is dissociated into OH adsorbed on the Mn

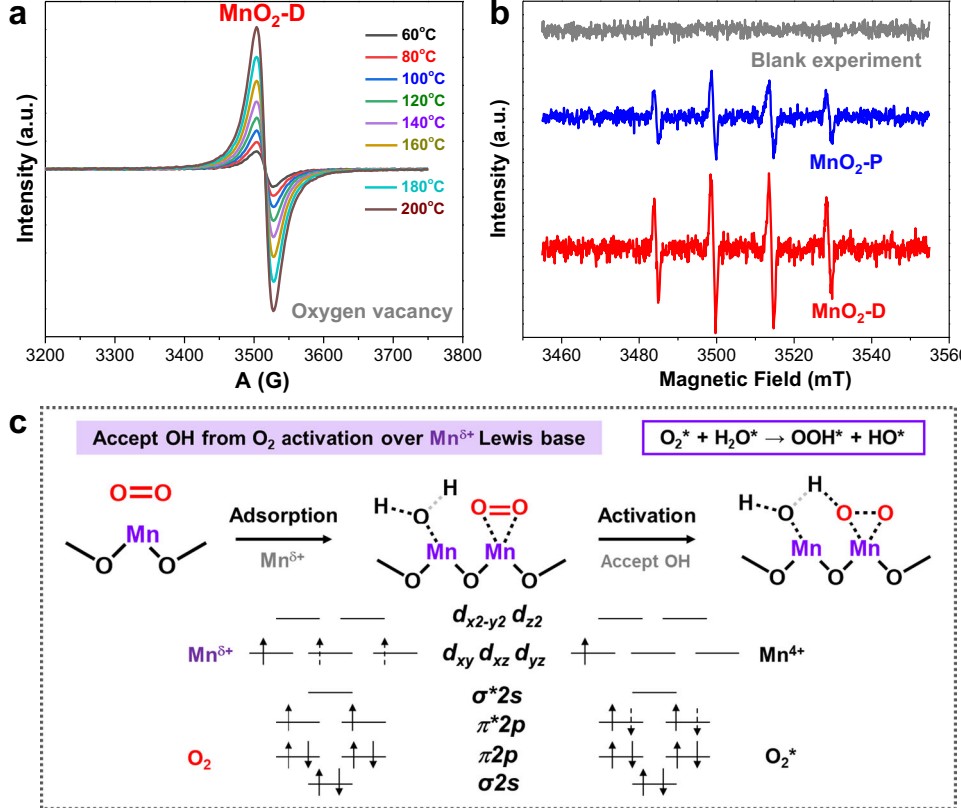

**Fig. 4 | Oxygen activation into hydroxyl radical on Lewis acid site in MnO$_2$-D.** **a** In situ EPR spectra for the detection of the evolution of oxygen vacancies on MnO$_2$-D. **b** In situ EPR spectra with free radical trapping agent (DMPO) for the oxidation of glycerol on MnO$_2$-P and MnO$_2$-D (for the capitation of hydroxyl radical). **c** Electron transition in orbit during O$_2$ adsorption over Mn$^{\delta+}$–O$_V$ structure.

center and H adsorbed on adjacent O$_{Mn}$ or O$_V$. The latter process well resembles the traditional definition of Lewis acid-base, where one accepts OH and the other accepts H. Finally, the Mn$^{\delta+}$ Lewis acid could easily accept the as-formed OH to formation Mn$^{\delta+}$–OH. Isotope experiments were further conducted to investigate whether lattice oxygen in MnO$_2$-D participates in the oxidation reaction (Mars van Krevelen (MvK) mechanism). Supplementary Fig. 9 shows that labeled $^{18}$O in MnO$_2$-D is not observed in the oxidation products of glycerol. This confirms that the liquid-phase oxidation reaction on MnO$_2$-D does not occur via the MvK mechanism. Rather, O$_2$ and H$_2$O dissociate into hydroxyl radical on the defective Mn$^{\delta+}$–O$_V$ pairs, before participating in the oxidation reaction.

## Enhanced C–C and C–H bond activation over Lewis base sites in MnO$_2$-D

The reaction mechanism of polyol oxidation to formic acid over the Mn$^{\delta+}$–O$_V$ pair was further investigated by in situ Fourier transform infrared reflection (in situ FTIR), DFT calculation and reaction kinetic studies. Figure 5a shows that the α interaction (1125–1075 cm$^{-1}$) and γ interaction (1075–1000 cm$^{-1}$), attributable to alkoxy bond between primary hydroxyl and metal oxides, gradually increase. ρ(OH) at 1370 cm$^{-1}$ and δ(OH) at 1440 cm$^{-1}$ belonged to glyceraldehyde gradually increase, suggesting the activation of primary hydroxyl group in glycerol[71,74]. Meanwhile, the β interaction at 1200–1125 cm$^{-1}$, and ω(CH$_2$) at 1250 cm$^{-1}$ and τ(CH$_2$) at 1300 cm$^{-1}$ belonging to dihydroxyacetone are also enhanced. These suggest MnO$_2$-P activates primary and secondary hydroxyl groups indiscriminately. Compared with MnO$_2$-P, τ(CH$_2$), ω(CH$_2$) and the β interaction corresponding to the activation of secondary hydroxyl groups are significantly lower on MnO$_2$-D, thus the preferential activation of primary hydroxyl group and C–C bond cleavage promote the generation of formic acid.

It is found that MnO$_2$-D exhibits low activation energies for continuous C–C bond cleavage (Supplementary Fig. 10), but high activation energies for deep oxidation reactions (such as the dihydroxy oxidation and decarboxylation to CO$_2$). In contrast, the activation energies of C–H and C–C bonds activation on MnO$_2$-P is significantly higher than that on MnO$_2$-D, and furthermore, the activation energies of multiple C–C bond cleavage are close to that of decarboxylation to CO$_2$ on MnO$_2$-P. These corroborate in situ FTIR findings that MnO$_2$-P without Mn$^{\delta+}$–O$_V$ pairs does not activate glycerol effectively and that the specific selectivity towards formic acid is low.

Figure 5b presents a schematic of the oxidation mechanism of glycerol to FA via MnO$_2$-D. Initially, the Mn$^{\delta+}$–O$_V$ structure utilizes Mn$^{\delta+}$ to adsorb oxygen from the O–H bond, while the O$_V$ attracts hydrogen from the O–H bond, thereby achieving the dehydrogenation of the O–H bond. In the subsequent step, the synergistic adsorption of oxygen-containing intermediates at the Mn$^{\delta+}$ metal site assists O$_V$ in attracting hydrogen from the C–H bond, leading to the formation of glyceraldehyde. Under the attack of hydroxyl radical, the as-formed glyceraldehyde is rapidly transformed into CH$_2$OHCHOHCHOOH, which undergoes C–C bond cleavage over Mn$^{\delta+}$–O$_V$ structure. After a series of processes compromising multiple O–H bond dehydrogenations, C–H bond dehydrogenation, free radical attack, and C–C bond cleavage steps mentioned above, the original glycerol is converted into three molecules of FA product. Notably, detailed DFT calculation (Supplementary Figs. 11 and 12) shows that the C–H bond activation in the C$_1$H$_3$O$_2$* (HCHOOH*) intermediate with the highest free energy is the rate-determining step (RDS), and the activation energies of two C–C bond cleavages on MnO$_2$-D (Tri) is less than 1 eV. As compared to MnO$_2$-P (Penta) [ε$_d$ = 2.63 eV], the d-band center of MnO$_2$-D (Tri) [ε$_d$ = 2.91 eV] moves away from Fermi energy E$_F$, suggesting the electrons in the Mn 3$d$ orbital could effectively overflow into nearby O

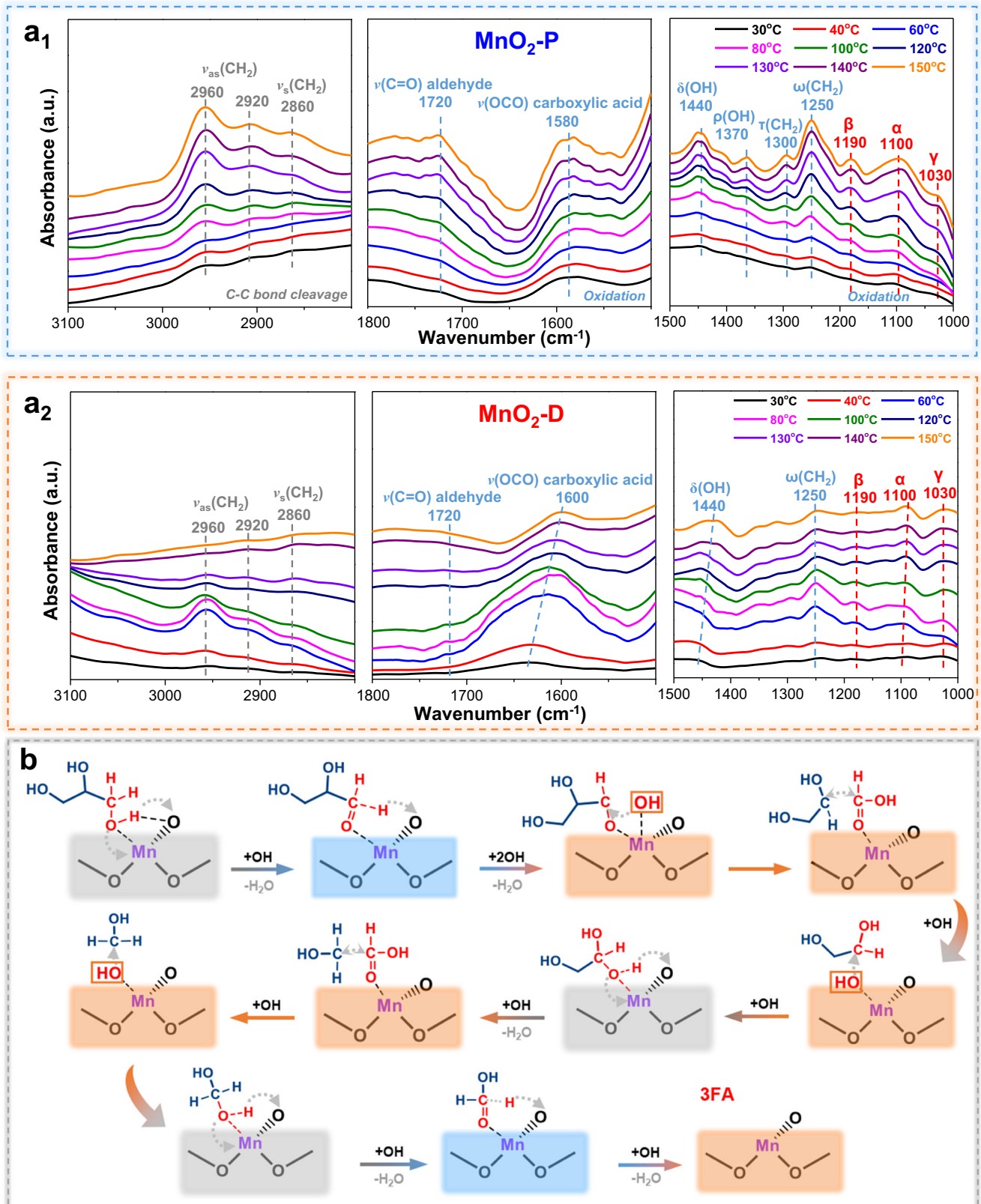

**Fig. 5 | Mechanistic investigations for cascade oxidation of polyol/sugar to formic acid. a** In situ FT-IR spectra of glycerol oxidation to formic acid over **a₁** MnO₂-P and **a₂** MnO₂-D. **b** Schematic diagram of reaction mechanism for the oxidation of glycerol to formic acid (the gray box represents O-H bond activation, the blue box represents C-H bond activation, and the orange box represents hydroxyl radical reaction and C-C bond cleavage).

(Supplementary Fig. 11c)[75,76]. Benefiting from the enhancement of O electronegativity (Lewis base) in the $Mn^{\delta+}$–$O_V$ pair, the binding ability of C and H is abnormally promoted. This provides a superior stable base for C–C bond cleavage and C–H bond activation, resulting in the low activation energies of C–C cleavage and C–H bond activation on $MnO_2$-D (Tri). Combined with micro-kinetic analysis (Supplementary Fig. 11d), the $MnO_2$-D (Tri) is located in the red region (i.e., high TOF) with ~8.0 eV of $E_C$ and ~6.0 eV of $E_O$, close to several noble metal catalysts[77].

## Discussion

In summary, we developed a defective α-$MnO_2$ catalyst enriched with Frustrated Lewis Pairs (FLPs) to promote the cascade oxidation of various polyols and sugars to formic acid. The reduction of oxygen coordination generates abundant exposed Mn species as Lewis acid sites, and strengthens the electron donating properties of adjacent oxygen to serve as Lewis base sites. During catalytic tests, a positive correlation between the abundance of strength-intensified FLPs in the defective $Mn^{\delta+}$–$O_V$ structure and catalytic activity was identified. Further characterization by various techniques provide evidence that the FLPs on defective $MnO_2$-D catalyst promote the $O_2$ as well as C–H and C–C bond activation synergistically on adjacent Lewis acid and base sites. As a result, the low-coordination $MnO_2$-D exhibits superior catalytic activity (TOF: 113.5 $h^{-1}$) and formic acid yield (formic acid yield >80%) for glycerol oxidation, which are comparable to the performance of previously reported noble metal catalysts. The catalyst is also effective in converting ethylene glycol, 1,2-propanediol, 1,3-propanediol, erythritol, xylitol and sorbitol into formic acid in yields ranging from 51.5 to 94.8%. This work demonstrates that metal oxide catalysts with water tolerant FLPs are promising for the oxidative polyol/sugar transformation.

## Methods

### Catalyst preparation

The $MnO_2$-P catalyst was prepared by a hydrothermal method[50]. In a typical process, 0.1 mol HCl and 0.25 mol $KMnO_4$ were added into 90 mL deionized water. Then, the mixed solution was transferred into a Teflon-lined stainless autoclave, which was hydrothermally treated at 140 °C (12 h). The as-formed precipitate was centrifuged and washed to remove excess ions. After drying in air overnight (100 °C), the sample was calcined to obtain the $MnO_2$-P catalyst in muffle furnace at 400 °C for 1 h.

The $MnO_2$-D catalyst was prepared by another hydrothermal method[50]. In a typical process, 1.58 g $KMnO_4$ was added in 30 mL deionized water, and 20 mL $(NH_4)_2C_2O_4 \cdot H_2O$ was further added in the above solution drop by drop. After 1 h of stirring, the mixture was transferred to a Teflon-lined stainless autoclave for 24 h at 180 °C. The as-formed powder was washed and filtered repeatedly. After drying in air overnight (100 °C), it was calcined to obtain the $MnO_2$-D catalyst in muffle furnace at 400 °C for 1 h.

### Catalytic test

Catalytic oxidation of polyol/sugar was carried out in the 50 mL autoclave. In a typical process, 25 mL deionized, 0.5 g substrate and 0.1 g catalyst were added in the autoclave. Through several times of $O_2$ washing, the final pressure was maintained at 1.0 MPa. After this oxidation reaction, the liquid product was analysed by the high performance liquid chromatography (HPLC) equipped with refractive index (RID-10A) and UV detectors (Shimadzu LC-20AT). The Rezex ROA-Organic Acid H+ (8%) was used as the chromatographic column in the mobile phase (0.005 M $H_2SO_4$). The gas product was detected by the Chromatography equipped with a FID and TCD detector (Scion 456-GC). The definitions of conversion (X), product selectivity (S), turnover frequency (TOF) and carbon mass balance (CMB) were calculated by the following formula:

$$X(\%) = \frac{K_0 - K_t}{K_0} \times 100\%$$

$$S(\%) = \frac{C_t}{K_0 - K_t} \times 100\%$$

$$TOF = \frac{N}{M \times T}$$

$$CMB = \frac{\sum_{i=1}^{3} i \times moles\ of\ Ci\ product\ (including\ unconverted\ glycerol)}{3 \times moles\ of\ glycerol\ substrate} \times 100\%$$

### Theory calculations

The $Dmol^3$ in the Material Studio 8.0 was employed to perform the DFT calculation. The Perdew–Burke–Ernzerh of (PBE) functional with the generalized gradient approximation (GGA) was selected to determine the correlation energies. Sampling brillouin zone $(3 \times 3 \times 1)$ via Monkhorst–Pack method and the double numerical plus polarization (DNP) together with effective core potentials were used. The TS parameters for van der Waals dispersion correction was considered, and all the energies were corrected by zero point energy (ZPE). The completed LST/QST method was used to search the transition state. Allowable deviations for displacement, gradient and total energy are 0.005 Å, 0.002 Ha/Å and $1.0 \times 10^{-5}$ Ha, respectively. The cif document of the $MnO_2$-P and $MnO_2$-D catalysts were obtained from the refined XRD, and detailed information was provided in the supporting data. For the two models, the 211 and 310 surface was cut from the $MnO_2$ bulk crystal. The 3-layer $p(2 \times 2)$ $MnO_2$-P and 3-layer $p(2 \times 2)$ $MnO_2$-D were constructed to represent the $MnO_2$-P and $MnO_2$-D (T) catalysts respectively. During the geometry optimization, the bottom layer atoms of the two models were fixed and the other atoms were relaxed. The adsorption energy is defined as $E_{ads} = E_{MnO2+substrate} - E_{MnO2} - E_{substrate}$, where $E_{MnO2+substrate}$, $E_{MnO2}$ and $E_{substrate}$ are the total energy of the calculation system, the isolated energy of calculation model and substrate, respectively.

All energy correction terms are extracted from the normal mode analyses of transition state the optimized reactant at various temperatures. Gibbs free energies correction has contained the ZPE correction. Outputs from $Dmol^3$ calculations include corrections to consider bare Gibbs free energies and electronic energies. The activation barrier (ΔE), the activation barrier with zero-point vibrational energy correction ($ΔE_Z$) and free energy presented (ΔG) are obtained using the following formula:

$$\Delta E_z = \Delta E + \Delta ZPE \tag{1}$$

$$\Delta E = E_{TS} - E_R \tag{2}$$

$$\Delta ZPE = ZPE_{TS} - ZPE_R \tag{3}$$

$$\Delta G = \Delta E_z + \Delta G_{corr} \tag{4}$$

$$\Delta G_{corr} = G_{TS} - G_R \tag{5}$$

Where $E_R$ and $E_{TS}$ are the electronic ground state energies of the reactant and transition state, respectively. The $ZPE_{TS}$ and $ZPE_R$ are the respective zero point vibrational energy (ZPE) corrections. $G_R$ and $G_{TS}$ are the free energy corrections of the reactant and the transition state, respectively.

## Characterizations

Rietveld refinements were carried out using TOPAS Academic[51,53,59]. Firstly, Pawley fitting was performed to refine the lattice parameters. The background was modeled using the Chebyshev function with 12 parameters. The peak shape was modeled via the Stephens peak shape function (an approach to spherical harmonics for *hkl* dependent peak shapes) considering the strain anisotropy broadening[3]. The initial atomic coordinates for the Manganese oxides with space group I4/m (87) were generated from the crystal structure in the Pearson's Crystal Data (#1102438)[4]. The refined parameters such as lattice parameters, background and sample specimen displacement from the Pawley fitting were kept the same for the Rietveld refinements. The z atomic coordinates of the Oxygen and Manganese ions were set to 0. Then, the following parameters were refined in sequence: (1) scale factor, (2) O, Mn atomic coordinates (x and y), (3) atomic (Oxygen and Manganese ions) occupancies and (4) the overall atomic displacement parameters. All the Rietveld refinements gave satisfactory agreement factors.

Scanning electron microscopy (SEM) was used to obtain the morphology of the $MnO_2$-P and $MnO_2$-D catalysts on a Hitachi S-4800. High-angle annular dark-field scanning transmission electron microscopy (HAADF-STEM) was carried out on the Titan 80–300 scanning/transmission electron microscope. X-ray photoelectron spectroscopy (XPS) was measured on the Thermo ESCALAB 250Xi with the correction of C 1 s binding energy of 284.8 eV, and other testing and analysis details were provided in Supporting Information. $N_2$ physisorption was measured on Micromeritics ASAP 2020. X-ray absorption spectroscopy (XAS) were measured on the Advanced Photon Source at Argonne National Laboratory (fluorescence mode on beamline 12-BM). The ATHENA module was employed to deal the data of X-ray absorption near edge structure (XANES) and extended X-ray absorption fine structure spectroscopy (EXAFS). $CO_2$ temperature programmed desorption ($CO_2$-TPD), $H_2$-temperature programmed reduction ($H_2$-TPR), $NH_3$-TPD and $O_2$-TPD were measured on the Micromeritics AutoChem II 2920. For the $CO_2$-TPD, $O_2$-TPD and $NH_3$-TPD, 0.1 g sample was treated in the $N_2$ gas at 300 °C for 1 h (60 mL/min). After the cool down of the sample (50 °C), 5 vol% $O_2$ ($NH_3$ or $CO_2$) gas in 95 vol% $N_2$ was introduced for 1 h (30 mL/min). Then, the quartz tube was heated to 1000 °C at a rate of 20 °C/min. For $H_2$-TPR, the sample was reduced in 10 vol% $H_2$ in Ar (60 mL/min) from 50 °C to 1000 °C with a heating rate of 15 °C/min. All the signals were collected by TCD detector. UV–vis absorption spectras were measured on a UV–vis–NIR Cary 5 (Varian) spectrophotometer. NMR was conducted on a Bruker Ascend 400 MHz NMR spectrometer. Mass spectrometric analysis was conducted on a QMS 200 (Balzers) quadrupole mass spectrometer.

Electron paramagnetic resonance (EPR) spectra were measured to obtain oxygen vacancy on a Bruker EMX-6/1spectrometer at 298 K. For the in situ EPR experimental (detection of oxygen vacancy), an in situ cell was loaded by 10 mL mixed solution of glycerol and sodium hydroxide (0.05 mol/L). The system pressure drop was monitored in real time. The sample was heated using a Bruker EMX plus continuous flow temperature control system. During this process, pure $O_2$ (100 mL/min) was purged into the cell. The EPR spectra were collected between 2400 and 3600 G in 83 ms. The microwave frequency was 9.3 GHz with a power of 0.2 mW, and the field was modulated at 100 kHz and with an amplitude of 5 G. For the Operando EPR experimental (detection of hydroxyl radical), three kinds of contrast experiments were designed. Fully put $MnO_2$-P and $MnO_2$-D catalysts into the evaluation conditions for reaction (120 °C for 4 h), and immediately put the samples after the reaction with DMPO into EPR for testing. The other group is a parallel blank experiment. Except that no catalyst is added, the other processes are completely consistent.

In situ, Fourier transform infrared reflection (in situ FTIR) was measured on the Thermo Scientific Nicolet iS50 FT-IR. For the in situ CO and $CO_2$, the samples were pretreated by 50 mL/min of $N_2$ at 300 °C for 1 h. After the sample cools to 80 °C, 5 vol% CO or $CO_2$ in $N_2$ (40 mL/min) was purged for 20 min to realize the adsorption saturation. Then, 40 mL/min $N_2$ was performed to remove the gas phase CO or $CO_2$. The spectra were collected using 128 scans in a resolution of 4 cm$^{-1}$. Detailed testing processes of the in situ FTIR of glycerol oxidation were provided in Supporting Information.

## Data availability

All relevant data that support the findings of this study are presented in the manuscript and supplementary information file. Source data are available from the corresponding author upon reasonable request. Source data are provided with this paper.

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

## Acknowledgements

This work was supported by the National Natural Science Foundation of China (22108305, 22272199, and 21978325), the Natural Science Foundation of Shandong Province (ZR2021QB076, ZR2020YQ17, ZR2020KB006, ZR2023YQ009, and ZR2022MB015), Fundamental Research Funds for the Central Universities (20CX06073A) and Post-doctoral Research Foundation of China: China Postdoctoral international exchange program (2020072). N.Y. thanks the the National Research Foundation Singapore NRF Investigatorship (NRF-NRFI07–2021–0006) for the financial support.

## Author contributions

H.Y. and N.Y. conceived this work. H.Y. performed the experiments, collected the data, and wrote the paper. B.L. performed the XRD. X.Z., F.M., M.Z., Y.P., J.L., Y.W., and H.Z. conducted the evaluation experiment. Y.L. and X.F. helped with the characterization analysis. X.C., D.C., L.L., and H.S. assisted the XAS characterization and analysis. C.Y. and N.Y. helped with data analyses and discussions. N.Y. revised the paper. All authors contributed to editing the paper.

## Competing interests

The authors declare no competing interests.
