## [Peer Review File · Nature Communications]

Enhancing Polyol/Sugar Cascade Oxidation to Formic Acid with Defect Rich MnO₂ CatalystsREVIEWER COMMENTS

Reviewer #1 (Remarks to the Author):

Yan et al. have investigated whether MnO₂ is an effective catalyst for the aerobic oxidation of alcohols to formic acid and used glycerol as their primary substrate for the investigation. Specifically, the authors were interested in determining whether catalytic performance (i.e. conversion and reaction selectivity) was related to defects sites in the MnO₂ materials. To probe this, they used a variety of different characterization techniques and demonstrated that there were clear differences in the physicochemical properties of the MnO₂ materials that they were examining; a defective MnO₂ material (MnO₂-D) and a comparatively non-defective MnO₂ material (MnO₂-P). They conclude that defective sites in the MnO₂ facilitate the formation of frustrated Lewis acid pairs and suggest that this is the key descriptor in the performance of these materials.

The characterization data is quite compelling; the authors clearly demonstrate that the two MnO₂ catalysts possess different catalytic properties. However, I have several concerns relating to how the authors have correlated these properties to the different catalytic performance the materials exhibit. To me, there is insufficient evidence to suggest that enhanced formic acid production (and catalytic activity) is attributed to the proportion of frustrated Lewis Acid pairs present. Firstly, the authors only use a sample set of two catalysts (the MnO₂-D and MnO₂-P). While these are indeed well characterized materials, I don't think correlating the physicochemical properties of two catalysts with catalytic performance is sufficiently robust enough to warrant publication in this prestigious journal. Furthermore, I have a number of issues related to how the authors compare catalyst performance in their reactions (see comments below).

Given the extensive concerns I have, in its current state I do not deem this acceptable for publication in Nature Commun. However, should the authors complete the additional experiments and eradicate the concerns I have I think it would make for a very nice publication. The aerobic oxidation of alcohols over cheap mixed metal materials is exceptionally challenging and providing a descriptor for how this can be achieved would be extremely useful for subsequent catalyst development and optimization.

Major concerns

The authors attempt to align defectivity (and frustrated Lewis Acid sites) of MnO₂ with catalyst performance - why do they use such a small sample set? The characterization and testing of additional MnO₂ materials (with varying proportions of defect sites). If similar correlations are observed with a greater sample set, the conclusions would be far more compelling.

The authors report a very low carbon mass balance for the blank reaction (Table S6). A glycerol conversion of 1.0 % is reported in the absence of the catalyst, but simultaneously report a carbon mass balance of 71.8 %; where is all the missing carbon? This makes me worry about the accuracy of the analytical methods employed throughout the work.

The authors compare the performance of the MnO₂-D catalyst at full conversion with the MnO₂-P catalyst at conversion of 31.8%. This is problematic for several reasons:

- Glycerol oxidation can lead to the formation of several intermediates before arriving at formic acid; it is a sequential oxidation process. Thus, reaction selectivity cannot be compared at different levels of conversion. I strongly recommend that the authors run time online reactions so that the proportions of products/intermediates can be monitored at different levels of substrate conversion.
- The activity of MnO₂-D catalyst is not correct – this should be acquired from an initial rate measurement.
- The stability data presented in Figure 3c is not informative; the authors are comparing the formic acid yield for the MnO₂-D at full conversion... when they compare the formic acid yield of the MnO₂-P catalyst (at a lower conversion - 31%) they report that the formic acid yield drops significantly; from ca. 10% to < 2%. Stable – think not.
- The authors state that the specific surface area of the materials cannot be related to the differences in activity exhibited by the two catalysts. How can the authors conclude this if the surface area of the MnO₂-D sample is four times that of the MnO₂-P sample, but they compare the activity at conversions of 99.2 and 31.6%, respectively.

To me, there is insufficient evidence to support the authors claims on the reaction mechanism. They demonstrate with EPR that hydroxyl/peroxy radicals are produced in-situ, but how can the authors be sure that these are the active species? If they were to run a reaction in the presence of a radical scavenger, would they see a decrease in rate? My concern is that in the caption of Figure 3, the authors state that they run reactions in the presence of NaOH. Is this the case and if so, what is its role in the reaction?

The authors present little characterization on the used catalyst– I would be very interested to see whether the FLPs remain after the reaction. Are the Lewis Acid sites water tolerant?

The authors should support their NH₃-TPD experiments with pyridine DRIFTS. The former does not distinguish between Bronsted and Lewis acid sites.

Minor Concerns

A few concerns on the XPS and associated fitting, please can the authors include more detail on the fitting constraints, spectrometer resolution and calibration method they used. The lattice oxygen component appears to move around, which is somewhat unexpected. Having checked previous work (Appl. Surf. Sci. 257 (2011) 2717), I also have concerns about the fitting of Mn oxidation states.

The article is generally well written, but a few typos are present.

The authors conduct mechanistic in-situ FTIR studies. This is notoriously very difficult to achieve due to the extremely low vapor pressure of glycerol. The authors should include a detailed methodology on how they conducted these experiments. Are the conditions used for these experiments similar to those used under reaction conditions

The authors compare the performance of their MnO₂-P catalyst with others in the literature. I just question whether these are the best in class examples to compare their work against. Formic acid is not a typical target compound from glycerol. The authors should also include the different conditions used in each case (e.g. temperature, pressure)

Reviewer #2 (Remarks to the Author):

Yan et.al developed a defective MnO₂ catalyst (MnO₂-D) with intensified Frustrated Lewis Pairs (FLPs) through a coordination number reduction strategy to enhance the aerobic oxidation of various polyols/sugars to formic acid. The tri-coordinated Mn in the defective MnO₂ catalyst exhibits typical characteristics of FLPs, where oxygen vacancies create Mn^{δ+} Lewis acid sites that activate O₂ and nearby electron-reconstructed oxygen creates Lewis base sites to facilitate C–C and C–H bond activation. It is an interesting study. However, there are still some problems that need to be improved before consideration.

1. There are many reports about the solid FLP catalysts. However, the introduction is insufficient to introduce the state-of-the-art progress. Compared with the previous reports, what is the novelty of the method or strategy for constructing the FLP sites.
2. After the reaction, characterizations of the spent catalyst using XRD, UV-Vis and XPS techniques prove that the defective Mn^{δ+}–O_v structure in MnO-D was well-preserved during the reaction. Why the MnO₂-D catalysts exhibited the 10% drop during the stability test. Meanwhile, the catalytic activity of MnO₂-D was fully restored. The author is suggested to explain the phenomenon.
3. The TOF value. The active sites on MnO-D were determined by ethanol adsorption. How does this TOF value compare to other reported results?
4. The EPR results. In Figure 4b, the quantitative relation is suggested to further clarified.

Reviewer #3 (Remarks to the Author):

The manuscript from Yan et al explores the use of defective manganese oxide to catalyse the oxidation of a range of polyols to formic acid. The MnO₂ with more oxygen vacancies (MnO₂-D) has an enhanced catalytic performance compared to the nominally pristine MnO₂ sample (MnO₂-P). This discovery is in line with a large number of reports on the role of defects in promoting heterogeneous catalysis (see recent review for many examples: <https://pubs.acs.org/doi/pdf/10.1021/acscatal.0c03034>).

The article is well written and clearly laid out. The materials appear to be well characterised, and I believe enough experimental details are provided for other researchers to replicate these results. The evidence clearly shows the MnO₂-D is a more efficient catalyst than MnO₂-P (Figure 3a). The recycling experiments and conversion of postulated intermediates to formic acid provides nice mechanistic information, and the control experiments have been well done; for example, the data in Supplementary

Figure 5 to rule out the importance of specific surface area between the two catalysts.

I do question the validity of invoking the concept of frustrated Lewis pairs (FLPs) to describe this work and mechanism. There are examples in the literature of using FLPs to describe the reactivity at surfaces in heterogeneous catalysis; some of these the authors have cited, but there are key references missing for this topic:

- FLPs on the surface of alumina (<https://onlinelibrary.wiley.com/doi/10.1002/anie.201006794> and <https://pubs.acs.org/doi/10.1021/ja3042383>) these articles are the first that relate reactivity at vacancies to FLPs

- There has also been extensive work on indium oxide surfaces, typified by this article:

<https://pubs.rsc.org/en/content/articlelanding/2015/CP/C5CP02613J>

However, in these examples the surfaces do actually behave somewhat like FLPs; i.e. you have a Lewis acidic site and Lewis basic site that can carry out cooperative reactivity on a small molecule. In this current manuscript from Yan et al, there are acidic sites (Mn) and basic sites (O), and the authors show very nicely that the relative acidity and basicity of these sites is enhanced in MnO₂-D (Figure 2g-1 and 2g-2). However, the reaction that is then catalysed (oxidation of glycerol) bears little resemblance to a reaction that goes via Lewis acidic and basic sites; instead the first step in Figure 4c shows two adjacent Mn centres activating O₂ and water to give a hydroxyl radical and the “Lewis base” is not involved at all. The ensuing cascade mainly involves the generation and transfer of hydroxyl radicals to glycerol, as stated at the bottom of page 11: “O₂ mainly reacts with H₂O to generate hydroxyl radical to participate in the subsequent activation of glycerol, which is consistent with previous reports [ref 47]” While there is growing interest in the use of radicals in FLP chemistry, this proposed mechanism does not correlate to any of those, as the reactivity is just at the Mn centres (which are more reactive in MnO₂-D due to being low-coordinate relative to MnO₂-P).

The dehydrogenation steps are the closest to FLP reactivity (e.g. from R1 to P3, where H₂ is lost from the starting glycerol, or R7 to P8), but even here the first H is removed as H-radical by a hydroxyl radical (not by either component of the “FLP”), and then there is a second homolytic cleavage where another H-radical is transferred to an oxygen in the catalyst, so this isn’t typical FLP reactivity either.

It must be said that the proposed mechanism is quite challenging to follow and unnecessarily time-consuming to understand properly. Figure 5 in the manuscript should contain a schematic of the mechanism instead of the strings of text underneath the energy diagrams in section b. Furthermore, there are a host of errors or confusing details in the configuration diagrams in Supplementary Figure 11:

- R1 and P1 in MnO₂-D are the same image, clearly R1 is wrong

- R2 and P2 in MnO₂-D are the wrong way round

- P(n) and R(n+1) should be the same for the energy diagram in Figure 5 to make any sense. For MnO₂-D, P1 has the central OH bond activated, but then in R2 (notwithstanding R2 and P2 are the wrong way round), it is one of the terminal OH groups that is activated, and it must be the latter case for the subsequent reactivity so P1 is presumably wrong?

- For MnO₂-P, R4 is wrong and there has been migration of an H atom from C to O (compared to P3 and

the formula above the panel, and compared to the same species in MnO₂-D. TS4 and P4 are also mislabelled as TS10 and P10

- Furthermore, R4 in MnO₂-P is also missing the hydroxyl group that is required for the attack
- For MnO₂-D, there is a large rearrangement between R4 and TS4. Can the authors please confirm that IRC calculations have been carried out on transition states to show that the R(n) and P(n) are actually linked by TS(n) in each case?
- The caption to Supplementary Figure 11-3 states that there is a “Mulliken charge distribution diagram of the rate-determining step (Step 10) in the MnO₂-P and MnO₂-D” – this appears to be missing (and there is no step 10 at all?)

Overall, I think the experimental part of the paper is very good and demonstrates that MnO₂-D is a good catalyst for the oxidation of glycerol, but the DFT mechanistic investigation needs to be sorted out before the paper can be published. Furthermore, I don't see what the authors have to gain by labelling this as “FLP” chemistry; it is a radical cascade mechanism that is promoted very well by the MnO₂ system (and outcompetes precious metal systems according to Figure 3b). That is not to say that this system couldn't behave as an FLP under different reaction conditions, as the authors have shown that there are accessible acidic and basic sites on the surface, but this reaction is not a good example to demonstrate this point.

Responses to Reviewers' Comments

Dear Reviewers,

We greatly appreciate the opportunity to revise our manuscript. The insightful suggestions and comments from the reviewers have deepened our understanding of defective metal oxide characterization and FLPs analysis. We've carefully considered all comments and supplemented the manuscript with additional experimental evidence to address the queries. We wish that these revisions will be satisfactory, and that the corrections will be well received.

Reviewer #1:

Yan et al. have investigated whether MnO_2 is an effective catalyst for the aerobic oxidation of alcohols to formic acid and used glycerol as their primary substrate for the investigation. Specifically, the authors were interested in determining whether catalytic performance (i.e. conversion and reaction selectivity) was related to defects sites in the MnO_2 materials. To probe this, they used a variety of different characterization techniques and demonstrated that there were clear differences in the physicochemical properties of the MnO_2 materials that they were examining; a defective MnO_2 material ($\text{MnO}_2\text{-D}$) and a comparatively non-defective MnO_2 material ($\text{MnO}_2\text{-P}$). They conclude that defective sites in the MnO_2 facilitate the formation of frustrated Lewis acid pairs and suggest that this is the key descriptor in the performance of these materials.

The characterization data is quite compelling; the authors clearly demonstrate that the two MnO_2 catalysts possess different catalytic properties. However, I have several concerns relating to how the authors have correlated these properties to the different catalytic performance the materials exhibit. To me, there is insufficient evidence to suggest that enhanced formic acid production (and catalytic activity) is attributed to the proportion of frustrated Lewis Acid pairs present. Firstly, the authors only use a sample set of two catalysts (the $\text{MnO}_2\text{-D}$ and $\text{MnO}_2\text{-P}$). While these are indeed well characterized materials, I don't think correlating the physicochemical properties of two catalysts with catalytic performance is sufficiently robust enough to warrant publication in this prestigious journal. Furthermore, I have a number of issues related to how the authors compare catalyst performance in their reactions (see comments below).

Given the extensive concerns I have, in its current state I do not deem this acceptable for publication in Nature Commun. However, should the authors complete the additional experiments and eradicate the concerns I have I think it would make for a very nice publication. The aerobic oxidation of alcohols over cheap mixed metal materials is exceptionally challenging and providing a descriptor for how this can be achieved would be extremely useful for subsequent catalyst development and optimization.

Major concerns:

1. The authors attempt to align defectivity (and frustrated Lewis Acid sites) of MnO₂ with catalyst performance - why do they use such a small sample set? The characterization and testing of additional MnO₂ materials (with varying proportions of defect sites). If similar correlations are observed with a greater sample set, the conclusions would be far more compelling.

Response:

Thanks for the reviewer's comment. We agree that a larger sample set would strengthen the reliability of the conclusion.

In response to the suggestion, we synthesized tetra-coordinated α -MnO₂ (MnO₂-T), by modifying the synthesis method (using potassium permanganate as precursor and manganese acetate as reducing agent). EXAFS in Supplementary Table 3-1 and Fig. R1a proves that the average coordination number of catalyst is 4.2. Together with MnO₂-P and MnO₂-D, we have a set of three MnO₂ samples with coordination number of 3.0 (MnO₂-D), 4.2 (MnO₂-T), and 5.2 (MnO₂-P), respectively. Under the same oxidation reaction conditions (120 °C, 6 h and 1MPa O₂), glycerol conversion reached 31.6% over MnO₂-P, 78.7% over MnO₂-T and 99.2% over MnO₂-D. The FA selectivity also increases with the decrease of coordination number in MnO₂.

To further substantiate our conclusion that low-coordination MnO₂ enhances glycerol oxidation, we prepared two additional phases, β -MnO₂ and γ -MnO₂, each of which exhibiting high and low coordination numbers respectively (four samples). By modulating the hydrothermal temperature or adapting the synthesis method, we successfully decreased the crystallinity of both catalysts, effectively reducing the coordination number of Mn. For β -MnO₂,

XRD in Fig. R2a confirms that both high-coordination (β -)MnO₂-H and low-coordination (β -)MnO₂-L are β crystal phase. EXAFS in Supplementary Table 3-2 and Fig. R1 proves that the coordination number of (β -)MnO₂-H and (β -)MnO₂-L are 5.4 and 3.4 respectively. The acidity and basicity of (β -)MnO₂-L with low coordination number are also enhanced, hinting at the presence of FLPs (Fig. R2b-c). Indeed, (β -)MnO₂-L exhibits higher glycerol conversion and FA selectivity than (β -)MnO₂-H (Supplementary Table 6-2). Similarly, for γ -MnO₂, (γ -)MnO₂-L exhibits higher glycerol conversion and FA selectivity than (γ -)MnO₂-H.

These new results corroborate our strategy of reducing the coordination number to facilitate the oxidation of polyols into formic acid, demonstrating its applicability across MnO₂ materials of varying crystal phases. We have integrated these results into the updated manuscript.

Supplementary Table 3-1. EXAFS fitting parameters at the Mn K-edge for MnO₂-P, MnO₂-T and MnO₂-D ($S_0^2=0.777$ for Mn)

Sample	Shell	CN ^a	R(\AA) ^b	$\sigma^2(\text{\AA}^2)$ ^c	$\Delta E_0(\text{eV})$ ^d	R factor
Mn foil	Mn-Mn	12*	2.66±0.01	0.007±0.002	6.69±2.67	0.016
	Mn-O	6	1.88±0.01	0.001±0.001	-3.41±1.68	
MnO ₂	Mn-Mn	4	2.87±0.01	0.002±0.0009	-8.07±1.31	0.02
	Mn-Mn	4	3.42±0.01	0.002±0.0009	-8.07±1.31	
	Mn-O	5.23±0.33	1.85±0.01	0.0013±0.001	0.77±1.45	
MnO ₂ -P	Mn-Mn	2.91±0.41	2.84±0.01	0.002±0.001	-4.28±1.88	0.015
	Mn-Mn	2.93±0.22	3.38±0.01	0.002±0.001	-4.28±1.88	
	Mn-O	4.20±0.51	1.89±0.01	0.00173±0.001	1.87±0.31	
MnO ₂ -T	Mn-Mn	2.86±0.34	2.99±0.01	0.00319±0.001	-6.61±0.78	0.006
	Mn-Mn	2.36±0.33	3.40±0.01	0.00121±0.001	-3.98±1.40	
	Mn-O	3.04±0.39	1.89±0.01	0.0012±0.0015	-0.48±1.84	
MnO ₂ -D	Mn-Mn	1.12±0.28	2.87±0.01	0.0005±0.0018	-0.06±0.01	0.012
	Mn-Mn	1.36±0.33	3.41±0.01	0.0005±0.0018	-0.06±0.01	

^aCN, coordination number; ^bR, distance between absorber and backscatter atoms; ^c σ^2 , Debye-Waller factor to account for both thermal and structural disorders; ^d ΔE_0 , inner potential correction; R factor indicates the goodness of the fit. S_0^2 was fixed to 0.777 for Mn, according to the experimental EXAFS fit of Mn foil by fixing CN as the known crystallographic value.

Supplementary Table 3-2. EXAFS fitting parameters at the Mn K-edge for (β -)MnO₂-H and (β -)MnO₂-H ($S_0^2=0.777$ for Mn)

Sample	Shell	CN ^a	R(\AA) ^b	$\sigma^2(\text{\AA}^2)$ ^c	$\Delta E_0(\text{eV})$ ^d	R factor
Mn foil	Mn-Mn	12*	2.66±0.01	0.007±0.002	6.69±2.67	0.016

	Mn-O	6	1.88±0.01	0.001±0.001	-3.41±1.68	
MnO ₂	Mn-Mn	4	2.87±0.01	0.002±0.0009	-8.07±1.31	0.02
	Mn-Mn	4	3.42±0.01	0.002±0.0009	-8.07±1.31	
(β-)MnO ₂ -H	Mn-O	5.35±0.80	1.88±0.01	0.00115±0.001	2.17±0.36	
	Mn-Mn	3.04±0.41	2.82±0.01	0.00316±0.002	-5.88±0.26	0.006
	Mn-Mn	1.65±0.32	3.43±0.01	0.00168±0.001	-2.16±1.94	
(β-)MnO ₂ -L	Mn-O	3.39±0.46	1.88±0.01	0.00129±0.001	1.97±0.21	
	Mn-Mn	2.14±0.89	2.84±0.01	0.00464±0.001	-5.02±0.12	0.006
	Mn-Mn	3.35±0.56	3.43±0.01	0.00198±0.0001	-2.36±0.96	

^aCN, coordination number; ^bR, distance between absorber and backscatter atoms; ^cσ², Debye-Waller factor to account for both thermal and structural disorders; ^dΔE₀, inner potential correction; R factor indicates the goodness of the fit. S0² was fixed to 0.777 for Mn, according to the experimental EXAFS fit of Mn foil by fixing CN as the known crystallographic value.

Supplementary Table 6-2. Oxidation of glycerol over MnO₂ catalysts

Catalyst	Selectivity (%)					Conversion (%)	Reaction conditions
	GLYA	GLYOA	OA	FA	Others		
(α-)MnO ₂ -P	17.4	30.6	19.5	24.1	8.1	31.6	120°C 6h
(α-)MnO ₂ -T	3.8	8.2	10.7	67.5	8.8	78.7	120°C 6h
(α-)MnO ₂ -D	1.4	3.2	2.8	83.2	8.8	99.2	120°C 6h
(β-)MnO ₂ -H	22.5	11.8	5.8	27.1	30.2	39.1	120°C 6h
(β-)MnO ₂ -L	10.5	5.7	2.5	63.5	15.9	79.0	120°C 6h
(γ-)MnO ₂ -H	15.2	9.1	10.6	26.2	37.1	41.4	120°C 6h
(γ-)MnO ₂ -L	12.5	0.0	5.3	56.5	23.9	80.7	120°C 6h
Only Glycerol	Glycerol conversion: 23.8%					-	120°C 0.5h
Methanol/Glycerol=0.5	Glycerol conversion: 8.9%			Methanol conversion: 34.4%			120°C 0.5h
Isopropanol/Glycerol=0.5	Glycerol conversion: 14.7%			Isopropanol conversion: 54.8%			120°C 0.5h
Tert butanol/Glycerol=0.5	Glycerol conversion: 11.5%			Tert butanol conversion: 20.0%			120°C 0.5h
MnO ₂ -D(base free)-1	0.5	4.1	2.0	68.9	23.1	29.1	120°C 6h
MnO ₂ -D(base free)-2	0.9	2.3	3.8	61.1	29.5	30.5	120°C 6h

Note: other reaction conditions: 0.1 g catalyst, 25 mL aqueous solution of glycerol (0.1 M), 0.5g NaOH, 1 MPa O₂.

Figure R1. Fourier transform of Mn K-edge extended EXAFS oscillations and the first-order derivatives of Mn K-edge XANES of (a) α - MnO_2 -T and (b) β - MnO_2 with high or low coordination number.

Figure R2. (a) XRD patterns, (b) NH₃-TPD and (c) CO₂-TPD of α-MnO₂, β-MnO₂ and γ-MnO₂ with high or low coordination number.

“Following exhaustive characterizations of MnO₂-D and MnO₂-P, we proceeded to synthesize tetra-coordinated α-MnO₂ (MnO₂-T) with an intermediate coordination number of 4.2 (Supplementary Fig. 2 and Supplementary Table 3). In addition, we synthesized and characterized two sets of distinct phases, β-MnO₂ and γ-MnO₂. Each set comprises a high coordination and a low coordination sample, respectively (Supplementary Table 3 and Supplementary Fig. 2). The intention behind these preparations was to establish a comprehensive comparative study on glycerol oxidation efficiency across various MnO₂ samples bearing different coordination environment.”[Page 6-7, Line 129-135]

“The tetra-coordinated MnO₂-T, characterized by its intermediate coordination number, exhibited moderate activity that fell between the performances of MnO₂-P and MnO₂-D catalysts (20.7 mmol/h/g_{cat}). Consistently, low-coordination number β-MnO₂ and γ-MnO₂ samples displayed superior catalytic performance in oxidizing glycerol to FA compared to their high coordination number counterparts (Supplementary Table 6). The above results indicates that the saturated coordination structure of MnO₂ does not have sufficient activity in C–H and C–C bond activation, implying that an unsaturated structure is key for high activity.”[Page 9-10, Line 189-195]

“Synthesis methods of other MnO₂ materials

Tetra-coordinated α-MnO₂ (MnO₂-T)

The KMnO₄ (0.79 g) and (CH₃COO)₂Mn·4H₂O (1.8 g) were dissolved in 50 mL of deionized water and stirred for 30 min. The solution was transferred in an autoclave, sealed, and placed in an oven at 140 °C for 2 h. After cooling to room temperature, the residue was washed with water thrice and dried at 60 °C in an oven. Finally, the sample was calcined to obtain the MnO₂-T catalyst in muffle furnace at 400 °C for 1 h.

β-MnO₂ [(β-)MnO₂-H and (β-)MnO₂-L]

The (NH₄)₂SO₈ (19.494 g) and MnCl₄·4H₂O (17.019 g) were mixed in 400 mL water and treated in the Teflon-sealed autoclave at 160°C for 12 h. The as-formed sample was calcinated at 400°C for 4h to obtain (β-)MnO₂-H. For the (β-)MnO₂-L, the MnCl₄·4H₂O and KMnO₄ (total amount of 20 mmol/L) was stirred for 2 h and then transferred into the Teflon-sealed autoclave at 160°C for 12 h..

γ-MnO₂ [(γ-)MnO₂-H and (γ-)MnO₂-L]

The (NH₄)₂SO₈ (9.006 g) and MnCl₄·4H₂O (7.916 g) were mixed in 400 mL water and treated in the Teflon-sealed autoclave at 90°C for 24 h [(γ-)MnO₂-H]. For the (γ-)MnO₂-L, it was purchased from McLean reagent.”[Supporting Information]

2. The authors report a very low carbon mass balance for the blank reaction (Table S6). A glycerol conversion of 1.0 % is reported in the absence of the catalyst, but simultaneously report a carbon mass balance of 71.8 %; where is all the missing carbon? This makes me worry about the accuracy of the analytical methods employed throughout the work.

Response:

The poor carbon balance at 1% conversion is because of the much larger error in product analysis at very low concentrations of products. The carbon balance in our manuscript refers to the number of carbon moles of product divided by the number of carbon moles of converted glycerol, following the formula:

$$\text{Carbon balance} = \frac{\sum_{i=1}^3 i \times \text{moles of } C_i \text{ product}}{3 \times \text{moles of glycerol converted}} \times 100\%$$

In other words, 70% of the carbon balance is relative to 1% of the converted glycerol. If based on the overall feeding, the actual carbon balance exceeds 99% (99% unconverted glycerol + 0.7% detected product). Additional explanations have been added to the manuscript.

“The definitions of conversion (X), product selectivity (S), turnover frequency (TOF) and carbon balance were calculated by the following formula:

$$\text{Carbon balance} = \frac{\sum_{i=1}^3 i \times \text{moles of } C_i \text{ product}}{3 \times \text{moles of glycerol converted}} \times 100\%$$

”[Page 17, Line 339-343]

3. The authors compare the performance of the MnO₂-D catalyst at full conversion with the MnO₂-P catalyst at conversion of 31.8%. This is problematic for several reasons:

-(A) Glycerol oxidation can lead to the formation of several intermediates before arriving at formic acid; it is a sequential oxidation process. Thus, reaction selectivity cannot be compared at different levels of conversion. I strongly recommend that the authors run time online reactions so that the proportions of products/intermediates can be monitored at different levels of substrate conversion.

-(B) The activity of MnO₂-D catalyst is not correct - this should be acquired from an initial rate measurement.

-(C) The stability data presented in Figure 3c is not informative; the authors are comparing the formic acid yield for the MnO₂-D at full conversion... when they compare the formic acid yield of the MnO₂-P catalyst (at a lower conversion - 31%) they report that the formic acid yield drops significantly; from ca. 10% to < 2%. Stable - think not.

-(D) The authors state that the specific surface area of the materials cannot be related to the differences in activity exhibited by the two catalysts. How can the authors conclude this if the

surface area of the MnO₂-D sample is four times that of the MnO₂-P sample, but they compare the activity at conversions of 99.2 and 31.6%, respectively.

Response:

(A) Based on the suggestion, we first supplemented the reaction time profiles of the two catalysts. Fig. R3 shows that MnO₂-P achieved 63.8% conversion after a long reaction time of 10 h, while MnO₂-D achieved nearly 100% conversion in just 6 hours. Simultaneously, a considerable variation in product distribution between the two catalysts is observed.

For MnO₂-D, the primary product in the initial stage is formic acid (FA), followed by glyceric acid (GLYA). As the reaction time extends, the selectivity for GLYA decreases, whereas the selectivity for FA first increases then decreases. The CO₂ selectivity rises with the extension of the reaction time, suggesting a reaction path over MnO₂-D likely following this sequence: primary hydroxyl oxidation and C-C bond cleavage of glycerol→GLYA→GLYOA→FA→CO₂.

On the other hand, for MnO₂-P, the main product in a short reaction time is glycolic acid (GLYOA), followed by GLYA and tartronic acid (TA). As the reaction time extends, the selectivity for GLYOA, GLYA, and TA markedly decreases, while oxalic acid (OA) selectivity increases. FA selectivity marginally increases with the extension of the reaction time (but always remains below 30%), and prolonged reaction time cannot enhance its selectivity. This indicates that the reaction pathway on MnO₂-P is likely to be: primary/secondary hydroxyl groups and C-C bonds of glycerol→GLYOA+GLYA+TA→OA→FA→CO₂.

It appears that MnO₂-P tends to unselectively activate hydroxyl groups and C–C bonds, resulting in uneven product distribution and low FA selectivity. These conclusions align with in situ FTIR results. Additionally, as per the reviewer's suggestion, we have revised Fig. 4a to control the glycerol conversion of MnO₂-P and MnO₂-D at a similar level (~64%) for product distribution comparison. The reaction time profiles, and related discussions have been supplemented in the revised manuscript.

(B) The calculation of the initial conversion rate and TOF is performed under controlled short reaction times (5 min for MnO₂-D and 15 min for MnO₂-P) at low conversions (<20%). To facilitate the reader's comprehension, we have added this information to the Supporting Information. For instance, the procedure of calculating the initial reaction rate for glycerol oxidation, using the MnO₂-D catalyst, is as the following:

$$\text{Initial reaction rate} = \frac{N_{\text{glycerol, converted}}}{m_{\text{Catalyst}} \cdot \text{Time}} = \frac{0.1 \times 0.025 \times 10.4\%}{0.1 \times 0.083} = 31.3 \text{ mmol/h/g}_{\text{cat}}$$

[conditions : 0.1 M glycerol solution (25 mL), 10.4% glycerol conversion, 0.083 h reaction time, 0.1 g catalyst]

(C) Indeed, Fig. R3c shows that both catalysts exhibit good stability. Our main purpose is to point out that the defective MnO₂-D could remain its stable structure during the reaction process. Therefore, in order to emphasize it more clearly, we have removed the stability results of MnO₂-P in Fig. 3c into the Supporting Information, and only shows the trend of the conversion and FA selectivity of MnO₂-D with the number of recycles. Meanwhile, in Fig. R3 we have provided the glycerol conversion and formic acid selectivity of MnO₂-P and MnO₂-D under cyclic testing conditions of 120 °C and 6 h. As the number of cycles increases, the glycerol conversion of both catalysts decrease by approximately 20%, while the FA selectivity does not change significantly, indicating that both catalytic materials exhibit similar stability.

(D) Our previous discussions in the MS may have been unclear, causing confusion. The specific surface area of MnO₂-D (104.2 m²/g) is approximately four times that of MnO₂-P (25.1 m²/g). Therefore, for activity comparison, the addition amounts of MnO₂-D and MnO₂-P were 0.1 g and 0.4 g, respectively. Initially, we compared the reaction performance of MnO₂-D and MnO₂-P over 6 h, yielding conversion rates of 99.2% and 85.5%, respectively. Subsequently, we investigated the initial reaction rates of MnO₂-D and MnO₂-P catalysts at short reaction times (5 min for both) and low conversions, resulting in rates of 0.301 and 0.091 mmol/h/m² (normalized by specific surface area), respectively. This suggests that the specific surface area is not the primary factor influencing catalytic performance. We have added the relevant reaction conditions and explanations to the Supporting Information for further clarification.

Figure R3. Catalytic performance of glycerol oxidation on (a) MnO_2-P and (b) MnO_2-D catalysts as a function of reaction time (experiment conditions: 25 mL aqueous phase solution (0.1 M), 0.5 g NaOH, 0.1 g catalyst, 1 MPa O_2 , 120°C; Glyceric acid, propanedioic acid, glycolic acid, oxalic acid and glycerol are abbreviated as GLYA, TA, GLYOA, OA and FA respectively.). (c) Catalytic stability of the MnO_2-P and MnO_2-D under multiple cycle test conditions.

Fig. 3: Cascade oxidation of polyol/sugar to formic acid over MnO₂-P and MnO₂-D. **a** Cascade oxidation of glycerol to formic acid over the MnO₂-P and MnO₂-D [25 mL aqueous phase solution (0.1 M), 0.5 g NaOH, 0.1 g catalyst, 1 MPa O₂, MnO₂-P for 10 h, MnO₂-D first for 2 h, MnO₂-D second for 6 h, 120 °C]. **b** Oxidation of glycerol to formic acid over MnO₂-D and other reported catalysts. **c** Catalytic stability of the MnO₂-P and MnO₂-D under multiple cycle test conditions. **d** Catalytic performance of the MnO₂-D for catalytic oxidation of different substrates (polyols/sugars) to formic acid [Formic acid (FA); Glycolic acid (GLYOA); Oxalic acid (OA); Glycerol (GLYA); Tartronic acid (TA); Ethylene glycol (EG); 1,2-propanediol (PG-2); 1,3-propanediol (PG-3); Erythritol (ET); Xylitol (XT); Sorbitol (ST)].

“Next, catalytic oxidation of glycerol was tested using the prepared Mn catalysts under 1 MPa O₂ at 120 °C. Fig. 3a and Supplementary Table 6 shows that at low conversion (<20%), the initial reaction rate reached 31.3 mmol/h/g_{cat}, almost 20 times higher than that of MnO₂-P. Moreover, at a similar glycerol conversion level (~64%), MnO₂-D exhibits a high FA selectivity of 83.3%, while MnO₂-P displays a total selectivity of 66.9% for various carboxylic acids, including glyceric acid (GLYA), oxalic acid (OA), glycolic acid (GLYOA) and FA. The tetra-coordinated MnO₂-T, characterized by its intermediate coordination number, exhibited moderate activity that fell between the performances of MnO₂-P and MnO₂-D catalysts (20.7

mmol/h/g_{cat}).” [Page 9, Line 184-191]

“Moreover, MnO₂-D exhibits excellent stability for glycerol oxidation, with almost no change in FA selectivity (>80%) and a slight decrease in glycerol conversion (<20%) after 6 catalytic cycles due to the coverage of some surface active sites by carboxylic acid products (Fig. 3c and Supplementary Fig. 5).” [Page 10, Line 203-206]

“To eliminate the influence of specific surface area on catalytic activity, the quantity of the MnO₂-P catalyst 4 times that of the MnO₂-D catalyst was added in the process of catalyst evaluation [Reaction conditions: 0.4 g MnO₂-P catalyst or 0.1 g MnO₂-D catalyst, 25 mL aqueous solution of glycerol (0.1 M), 120 °C, 1 MPa O₂, 6 h]. The conversion of the MnO₂-P and MnO₂-D catalysts are 85.5 and 99.2% respectively, and the selectivity of formic acid on the MnO₂-P catalyst is only 10.5%, far lower than that of the MnO₂-D catalyst (83.2%).

Additionally, we determined the initial reaction rates of MnO₂-D and MnO₂-P catalysts at short reaction times (5 min for both) and low conversion rates, yielding 0.301 and 0.091 mmol/h/m² (normalized by specific surface area), respectively. It is clear that the activity of the MnO₂-P catalyst is markedly lower than that of the MnO₂-D catalyst. This suggests that the specific surface area is not the primary factor influencing catalytic performance. Instead, the Mn-O active sites with varied coordination structures, formed by the MnO₂-P and MnO₂-D catalysts, appear to be the critical factor affecting the oxidation of polyols to formic acid.”

[Supporting Information]

4. To me, there is insufficient evidence to support the authors claims on the reaction mechanism. They demonstrate with EPR that hydroxyl/peroxy radicals are produced in-situ, but how can the authors be sure that these are the active species? If they were to run a reaction in the presence of a radical scavenger, would they see a decrease in rate? My concern is that in the caption of Figure 3, the authors state that they run reactions in the presence of NaOH. Is this the case and if so, what is its role in the reaction?

Response:

Prompted by the suggestion, we incorporated methanol, isopropanol, and tert-butanol as hydroxyl radical quenching agents in the oxidation process to reinforce the notion that MnO₂ materials follow a mechanism involving active hydroxyl species in the activation of glycerol.

As demonstrated in Supplementary Table 6-2, the introduction of these hydroxyl radical quenchers, particularly methanol, significantly decreased glycerol conversion by preferentially interacting with the generated hydroxyl radicals. This indirectly substantiates that the hydroxyl radical, formed by the activation of O₂ and H₂O, is involved in glycerol activation. In our previous works (Appl. Sur. Sci, 2019, 497, 143661, AIChE J, 2022. doi.org/10.1002/aic.17868 and Appl. Catal. B 2019, 259: 118070) and the study by Zope et al. (Science. 2010, 330, 74-78), DFT calculations have corroborated that O₂ and H₂O mainly dissociate into adsorbed M–OH* on metal surfaces such as Pt, Au, Ru, and participate in the subsequent oxidation of glycerol molecules.

The main role of NaOH is to polarize the metal surface (M–OH*) in the initial reaction stage. Supplementary Table 6-2 shows the evaluation experiments of MnO₂-D under base-free conditions. Although FA is still the main product, MnO₂-D needs to react for 6 hours to achieve a conversion rate of ~30% (equivalent to the result under alkaline conditions for 1 hour). This is mainly because the M–OH* generated by the dissociation of O₂ and H₂O needs to reach surface equilibrium to promote the activation of intermediate (for example, aldehydes are attacked by M–OH* to form carboxylic acid like intermediates) and consume H removed from C–H bond and O–H bond to re-activate the metal oxide sites, which has been confirmed by previous reports (Chem. Eng. Sci. 2019, 203, 228-236; Nature Comm. 2022, 13, 5467). In addition, the introduction of base also sustains formic acid in the form of sodium formate, avoiding its decomposition at above 100 °C.

Additional discussions have been added into the revised manuscript.

Supplementary Table 6-2. Oxidation of glycerol over MnO₂ catalysts

Catalyst	Selectivity (%)					Conversion (%)	Reaction conditions
	GLYA	GLYOA	OA	FA	Others		
(α -)MnO ₂ -P	17.4	30.6	19.5	24.1	8.1	31.6	120°C 6h
(α -)MnO ₂ -T	3.8	8.2	10.7	67.5	8.8	78.7	120°C 6h
(α -)MnO ₂ -D	1.4	3.2	2.8	83.2	8.8	99.2	120°C 6h
(β -)MnO ₂ -H	22.5	11.8	5.8	27.1	30.2	39.1	120°C 6h

$(\beta\text{-})\text{MnO}_2\text{-L}$	10.5	5.7	2.5	63.5	15.9	79.0	120°C 6h
$(\gamma\text{-})\text{MnO}_2\text{-H}$	15.2	9.1	10.6	26.2	37.1	41.4	120°C 6h
$(\gamma\text{-})\text{MnO}_2\text{-L}$	12.5	0.0	5.3	56.5	23.9	80.7	120°C 6h
Only Glycerol	Glycerol conversion: 23.8%			-			120°C 0.5h
Methanol/Glycerol=0.5	Glycerol conversion: 8.9%			Methanol conversion: 34.4%			120°C 0.5h
Isopropanol/Glycerol=0.5	Glycerol conversion: 14.7%			Isopropanol conversion: 54.8%			120°C 0.5h
Tert butanol/Glycerol=0.5	Glycerol conversion: 11.5%			Tert butanol conversion: 20.0%			120°C 0.5h
$\text{MnO}_2\text{-D}(\text{base free})\text{-1}$	0.5	4.1	2.0	68.9	23.1	29.1	120°C 6h
$\text{MnO}_2\text{-D}(\text{base free})\text{-2}$	0.9	2.3	3.8	61.1	29.5	30.5	120°C 6h

Note: other reaction conditions: 0.1 g catalyst, 25 mL aqueous solution of glycerol (0.1 M), 0.5g NaOH, 1 MPa O_2 .

“The hydroxyl radical quenching experiment, employing methanol, isopropanol, and tert-butanol, further established that the produced hydroxyl radical significantly impacts the oxidation reaction (Supplementary Table 6). This suggests that O_2 mainly interacts with H_2 to generate hydroxyl radicals that participate in the subsequent glycerol activation, aligning with previous studies⁶⁷.” [Page 12, Line 237-239]

5. The authors present little characterization on the used catalyst- I would be very interested to see whether the FLPs remain after the reaction. Are the Lewis Acid sites water tolerant? The authors should support their NH_3 -TPD experiments with pyridine DRIFTS. The former does not distinguish between Bronsted and Lewis acid sites.

Response:

We have supplemented the py-IR of both $\text{MnO}_2\text{-P}$ and $\text{MnO}_2\text{-D}$. Fig. R4 illustrates that both catalysts exhibit peaks at 1450 cm^{-1} , attributed to Lewis acid sites, with $\text{MnO}_2\text{-D}$ presenting a stronger peak. To assess the water tolerance of these Lewis acid sites, we tested $\text{MnO}_2\text{-P}$ and $\text{MnO}_2\text{-D}$ pre-treated with water under non-dehydrating conditions. The Lewis acid peak intensity of $\text{MnO}_2\text{-D}+\text{H}_2\text{O}$ at 1450 cm^{-1} remained virtually unchanged, while the Lewis acid peak of $\text{MnO}_2\text{-P}+\text{H}_2\text{O}$ exhibited a slight increase, indicating that the Lewis acid sites are water tolerant.

Meanwhile, we further supplemented the NH_3 -TPD and CO_2 -TPD of spent MnO_2 -D. Fig. R5 shows that the acid and basic properties of the spent MnO_2 -D (re-calcined) are not significantly different from those of the fresh MnO_2 -D. There is only a slight decrease in medium strong acid in spent MnO_2 -D due to the slight damage of partial structure under long-term reaction conditions. Moreover, MnO_2 -D still exhibits stronger and more acid-base sites than MnO_2 -P. On these foundations, it is suggested that the FLPs formed in low-coordination MnO_2 -D exhibit good catalytic stability in the aqueous oxidation reaction. Relevant discussions have been added into the revised manuscript.

Furthermore, we have conducted NH_3 -TPD and CO_2 -TPD analysis of spent MnO_2 -D catalyst. Fig. R5 demonstrates that the acid-base properties of the spent (re-calcined) MnO_2 -D do not significantly differ from those of the fresh MnO_2 -D, with only a minor decrease in medium-strong acid in the spent MnO_2 -D due to marginal structural damage under long-term reaction conditions. Moreover, MnO_2 -D still presents stronger and more abundant acid-base sites than MnO_2 -P. Thus, it can be concluded that the acid and base sites in low-coordination MnO_2 -D exhibit commendable catalytic stability during the aqueous oxidation reaction. These insights have been incorporated into the revised manuscript's discussion section.

Figure R4. Pyridine (py)-IR of (a) MnO_2 -P and MnO_2 -D with and without water.

Figure R5. (a) NH_3 -TPD and (b) CO_2 -TPD of spent MnO_2 -P and MnO_2 -D.

“In addition, pyridine infrared results demonstrate that the Lewis acid sites on MnO_2 -P and MnO_2 -D are water tolerant (Supplementary Fig. 5).”[Page 9, Line 179-181]

“Characterizations of the spent catalyst using XRD, UV-Vis, XPS, NH_3 -TPD and CO_2 -TPD techniques prove that the defective $\text{Mn}^{\delta+}$ - O_V structure and acid-base properties in MnO_2 -D was well-preserved during the reaction (Supplementary Fig. 5).”[Page 10, Line 206-208]

“Meanwhile, we further supplemented the NH_3 -TPD and CO_2 -TPD of spent MnO_2 -D. Fig.5e shows that the acid and basic properties of the spent MnO_2 -D (re-calcined) are not significantly different from those of the fresh MnO_2 -D. There is only a slight decrease in medium strong acid in spent MnO_2 -D due to the slight damage of partial structure under long-term reaction conditions. Moreover, MnO_2 -D still exhibits stronger and more acid-base sites than MnO_2 -P. On these foundations, it can be concluded that the acid and base sites formed in low-coordination MnO_2 -D exhibit good catalytic stability in the aqueous oxidation reaction.

Supplementary Fig. 5f shows that both MnO_2 -P and MnO_2 -D exhibit the peaks at 1450 cm^{-1} belonging to Lewis acid, and MnO_2 -D displays stronger L acid peak. Meanwhile, in order to investigate the water tolerant of L acid, MnO_2 -P and MnO_2 -D with pre-added water were tested under non dehydrated conditions. It is found that the intensity of the L acid peak of MnO_2 -D+ H_2O at 1450 cm^{-1} is almost unaffected, and even the L acid of MnO_2 -P+ H_2O is slightly enhanced. Thus, the Lewis acid sites in the MnO_2 -P and MnO_2 -D are water tolerant.”[Supporting Information]

Minor Concerns

6. A few concerns on the XPS and associated fitting, please can the authors include more detail on the fitting constraints, spectrometer resolution and calibration method they used. The lattice oxygen component appears to move around, which is somewhat unexpected. Having checked previous work (Appl. Surf. Sci. 257 (2011) 2717), I also have concerns about the fitting of Mn oxidation states.

Response:

X-ray photoelectron spectrometer (ESCALAB 250Xi, ThermoFisher, USA) was used to collect XPS data. The optimal spatial resolution, optimal energy resolution and (large area) energy resolution of monochromator are 20 microns, 0.45 eV and 400000 cps (FWHM \leq 0.5 eV), respectively. The excitation source was Al K α ray ($h\nu=1486.6$ eV), the working voltage was 12.5 kV, the filament current was 16 mA, the spot beam was 400 μm , and the signal accumulation was carried out for 10 cycles. The full spectrum of the test pass-energy is 100 eV, the narrow spectrum is 20 eV, the step size is 0.1 eV, and the residence time is 40-50 ms. Test steps: the powder is pressed and prepared into the sample plate, then put into the instrument for vacuum testing. Correction principle: the default is C1s=284.8 eV combined with energy standard for charge correction.

We have also added more details on the fitting constraints and calibration method. Several basic rules of XPS peak-splitting fitting are as follows:

1) The binding energy of different chemical state of each element reference online database at <http://srdata.nist.gov/xps/selectEnergyType.aspx>.

2) The splitting intensity ratio of p, d and f levels is certain, $p_{3/2}:p_{1/2}=2:1$; $d_{5/2}:d_{3/2}=3:2$, $f_{7/2}:f_{5/2}=4:3$. This rule was followed during peak fitting.

3) For the energy level with splitting (p, d, f), the distance between the two splitting orbitals is basically fixed. For example, the difference between Mn $p_{3/2}$ and Mn $p_{1/2}$ in the same state is about 11.25 eV. The distance between the two orbitals of energy level splitting will be different in specific chemical states.

4) For a certain element, the full width at half maxima (FWHM) of the valence peak of the same orbital in different samples should be similar; for the split orbit of the same element, FWHM should be as close as possible. Generally, FWHM is no more than 2.7 eV.

5) For the same instrument and instrument parameters, Gaussian-Lorentzian (GL) ratio of each data shall be consistent. Generally, GL ratio is about 20%.

There is no error in the O1s spectrum. Through deconvoluting the O 1s spectrum into three different peaks O_I, O_{II} and O_{III}, oxygen species at the surface of MnO₂ can be identified. The three different peaks O_I, O_{II} and O_{III} at BE = 529.7, 531.5, and 532.9 eV is usually ascribed to the lattice oxygen (O_I), oxygen vacancy (O_{II}), and adsorbed molecular water (O_{III}), respectively (J. Am. Chem. Soc. 2019, 141, 2, 890-900). MnO₂-D exhibits more low-valence Mn^{δ+} than MnO₂-P, and the EPR result indicates that the abundant oxygen vacancies on the surface of MnO₂-D processes more unpaired electrons. This electronic reconstruction causes O1s in MnO₂-D to shift towards low binding energy, compared with that in MnO₂-P.

We apologize for any confusion resulting from the previous erroneous XPS peak deconvolution analysis of Mn 2p_{3/2}. It is common in the literature that the asymmetrical Mn 2p_{3/2} XPS spectrum of each sample can be deconvoluted to four components at B.E. = 641.0, 642.0, 643.0, and 644.5 eV, which assign to the surface Mn²⁺, Mn³⁺, and Mn⁴⁺ species and the satellite of the Mn³⁺ species, respectively (Chem. Eng. J. 2020, 394, 124458). Therefore, we re-deconvoluted the Mn 2p_{3/2} XPS spectrum in Fig. 2b and revised the content of satellite peak, Mn²⁺, Mn³⁺ and Mn⁴⁺ species in Supplementary Table 4. This revision consolidates the previously misclassified Mn⁶⁺ and Mn⁷⁺ into the current satellite peak. Despite these modifications, our fundamental conclusion remains unchanged: MnO₂-D bear more low-valence Mn^{δ+} species. Relevant revisions have been added into the revised manuscript and Supporting Information.

“Consistent with O₂-TPD and EPR analysis, the XPS spectra of O 1s region in Fig. 2d and Supplementary Table 4 demonstrates that the MnO₂-D catalyst has the highest oxygen vacancy content (27.6% O_{II}) with lower binding energy.” [Page 7, Line 156-158]

“X-ray photoelectron spectroscopy (XPS) was measured on the Thermo ESCALAB 250Xi with the correction of C 1s binding energy of 284.8 eV. The optimal spatial resolution, optimal energy resolution and (large area) energy resolution of monochromator are 20 microns, 0.45 eV and 400000 cps (FWHM ≤ 0.5 eV), respectively. The excitation source was Al Kα ray (hν=1486.6 eV), the working voltage was 12.5 kV, the filament current was 16 mA, the spot beam was 400 μm, and the signal accumulation was carried out for 10 cycles. The full spectrum

of the test pass-energy is 100 eV, the narrow spectrum is 20 eV, the step size is 0.1 eV, and the residence time is 40-50 ms. Test steps: the powder is pressed and prepared into the sample plate, then put into the instrument for vacuum testing. Several basic rules of XPS peak-splitting fitting are as follows: 1) The binding energy of different chemical state of each element reference online database at <http://srdata.nist.gov/xps/selEnergyType.aspx>. 2) The splitting intensity ratio of p, d and f levels is certain, $p_{3/2}:p_{1/2}=2:1$; $d_{5/2}:d_{3/2}=3:2$, $f_{7/2}:f_{5/2}=4:3$. This rule should be followed during peak fitting. 3) For the energy level with splitting (p, d, f), the distance between the two splitting orbitals is basically fixed. For example, the difference between Mn $p_{3/2}$ and Mn $p_{1/2}$ in the same state is about 11.25 eV. The distance between the two orbitals of energy level splitting will be different in specific chemical states. 4) For a certain metal, the full width at half maxima (FWHM) of the valence peak of the same orbital in different samples should be similar; for the split orbit of the same element, FWHM should be as close to the same as possible. Generally, FWHM is no more than 2.7 eV. 5) For the same instrument and instrument parameters, Gaussian-Lorentzian (GL) ratio of each data shall be consistent. Generally, GL ratio is about 20%.”[Supporting Information]

Fig. 2: Strength-intensified Frustrated Lewis Pairs of $\text{MnO}_2\text{-D}$. *a* The first-order derivatives of Mn K-edge XANES. *b* Mn 2p XPS spectra of $\text{MnO}_2\text{-P}$ and $\text{MnO}_2\text{-D}$. *c* O_2 -TPD and *d* XPS spectra of O 1s of $\text{MnO}_2\text{-P}$ and $\text{MnO}_2\text{-D}$. *e* Aberration-corrected HAADF-STEM image of $\text{MnO}_2\text{-P}$ and $\text{MnO}_2\text{-D}$. *f* Structure diagram and Mulliken charge distribution of Frustrated Lewis Pair in $\text{MnO}_2\text{-P}$ and $\text{MnO}_2\text{-D}$. *g* NH_3 -TPD and CO_2 -TPD of $\text{MnO}_2\text{-P}$ and $\text{MnO}_2\text{-D}$.

7. The article is generally well written, but a few typos are present.

Response:

Thank you for the insightful suggestion.

The typos of manuscript were checked. The relevant content has been revised in the revised manuscript.

e.g. “microkinetic” is revised into “micro-kinetic”.

“spectras” is revised into “spectrum”.

8. The authors conduct mechanistic in-situ FTIR studies. This is notoriously very difficult to achieve due to the extremely low vapor pressure of glycerol. The authors should include a detailed methodology on how they conducted these experiments. Are the conditions used for these experiments similar to those used under reaction conditions.

Response:

Thanks for the reviewer’s insightful comment.

We fully agree that it is very difficult to conduct FTIR analysis for the reaction due to the low vapor pressure of glycerol (0.4 kPa at 20 °C). Our previous works (Angew. Chem. Int. Ed. 2022, e202116059; AIChE J. 2022; e17833) and this manuscript both adopted a strategy of freeze drying method to pre-adsorb glycerol in advance, which is also widely used in the literature to study glycerol oxidation using FTIR techniques (Nat. Commun. 2022, 13, 5467; Appl. Catal. B, 2021, 291, 120061; Appl. Catal. B, 2021, 298, 120634). The detailed experimental steps are as follows:

For the in situ FTIR of glycerol oxidation, the catalysts were impregnated with glycerol via an ex-situ wet impregnation method. Certain amount of catalysts, typically 300 mg, was added to a 5 mL mixed solution of glycerol and sodium hydroxide (0.5 mol/L) in order to pre-adsorb quantitative glycerol and NaOH. The resulting slurry was mixed in a shaker with continuous magnetic stirring for 4 h at room temperature. The pre-absorbed sample was collected after low-speed centrifuging then dried through vacuum freeze-drying for 24 h. After that, spectra were collected over the treated catalysts as backgrounds at 25°C for 1h (record background) in N₂ (30 mL/min). Subsequently, the spectra of glycerol pre-absorbed samples were recorded at room temperature to characterize the adsorption form, then 5 vol% O₂ and N₂ mixture gas (30 mL/min) was introduced in the IR cell with the increase of temperature (200 °C). The final spectrum was obtained by subtracting the spectrum of the sample from the adsorbed sample spectrum.

We have added more detailed experimental details into the Supporting Information.

“For the in situ FTIR of glycerol oxidation, the catalysts were impregnated with glycerol via an ex-situ wet impregnation method. Certain amount of catalysts, typically 300 mg, was

added to a 5 mL mixed solution of glycerol and sodium hydroxide (0.5 mol/L) in order to pre-absorb quantitative glycerol and NaOH. The resulting slurry was mixed in a shaker with continuous magnetic stirring for 4 h at room temperature. The pre-absorbed sample was collected after low speed centrifugalization then dried through vacuum freeze-drying for 24 h. After that, spectra were collected over the treated catalysts as backgrounds at 25°C for 1h (record background) in N₂ (30 mL/min). Subsequently, the spectra of glycerol pre-absorbed samples were recorded at room temperature to characterize the adsorption form, then 5 vol% O₂ and N₂ mixture gas (30 mL/min) was introduced in the IR cell with the increase of temperature (200 °C). The final spectrum was obtained by subtracting the spectrum of the sample from the adsorbed sample spectrum.”[Supporting Information]

9. The authors compare the performance of their MnO₂-P catalyst with others in the literature. I just question whether these are the best in class examples to compare their work against. Formic acid is not a typical target compound from glycerol. The authors should also include the different conditions used in each case (e.g. temperature, pressure)

Response:

We appreciate the reviewer’s suggestion to more effectively present a comparison between the catalytic performance of MnO₂-D and other reported catalysts for glycerol oxidation. To this end, we’ve updated Supplementary Table 6-2 with additional comparison catalysts, relevant reaction conditions (temperature, pressure, and time), and grouped the catalysts into three distinct categories: (1) noble metals + O₂ (mainly yielding other keto acids), (2) non-noble metals + O₂, and (3) heterogeneous catalysts + H₂O₂.

This classification is designed to underscore specific aspects of MnO₂-D’s performance: Category (1) serves to illustrate that the catalytic activity of MnO₂-D against that of precious metals in O₂-mediated glycerol oxidation. Category (2) emphasizes that among non-precious metal catalysts using O₂, MnO₂-D offers the highest catalytic activity and FA selectivity. Category (3) showcases MnO₂-D’s superior ability to activate O₂, outperforming heterogeneous catalysts when H₂O₂ serves as the oxidant.

We’ve incorporated these revisions and corresponding discussions in the Supporting Information.

Supplementary Table 6-2. Oxidation of glycerol over MnO₂ catalysts

	Catalyst	Selectivity (%)				Conversion (%)	TOF (h ⁻¹)	Reaction conditions	Reference (Year)
		GLYA	GLYOA	TA	FA				
MnO₂-D+O₂	MnO ₂ -D	1.4	3.2	0.0	83.2	99.2	113.5	120°C, 1MPa, 8 h	This work
Noble metals + O₂	Pt ₁ /HAP	90.3	1.8	7.8	0.0	91.2	351.2	50°C, 1MPa, 16 h	[1] (2022)
	AuCu/ZnO	3.6	1.1	-	-	3.1	75.4	60°C, 1MPa, 8 h	[2] (2022)
	Pt ₁ +Pt _n /Cu-CuZrO _x	80.0	1.1	1.4	0.0	90.1	224.0	60°C, 1bar, 8 h	[3] (2022)
	Pt/MgO/SAB-15 (0.1)	65.5	5.4	2.0	0.9	60.2	1671.2	60°C, 1MPa, 8 h	[4] (2019)
	Pt/SAB-15	49.9	5.0	2.0	0.1	25.5	121.0	60°C, 1MPa, 8 h	[4] (2019)
	Cu ₁ Mg ₄ O _x	8.9	60.3	0.1	22.0	89.2	0.2	200°C, 1MPa, 4 h	[5] (2022)
	Mn ₂ O ₃ -T	9.8	41.7	-	-	68.6	0.4	140°C, 1MPa, 12h	[6] (2021)
Non noble metals + O₂	Ni ₁ Co ₁ O _x	66.5	19.5	-	6.9	78.5	5.4	80°C, 1MPa, 24 h	[7] (2020)
	Co _{0.15} /Mg ₃ Al-c	29.2	-	46.9	13.4	1.1	0.7	70°C, 1MPa, 24 h	[8] (2016)
	Co _{0.15} /Mg ₃ Al-s	2.1	-	63.5	2.4	100	2.4	70°C, 1MPa, 24 h	[8] (2016)
	V ₃ /ZrP-m	-	-	-	50.9	89.0	0.7	170°C, 3MPa, 4 h	[9] (2019)
Heterogeneous catalysts +H₂O₂	CuN ₄ /NCSWCs	0.0	6.4	0.0	61.3	96.3	1.1	80°C, 8 h	[10] (2022)
	Au-PTA/MSN	-	-	-	79.2	33.6	7.1	80°C, 24 h	[11] (2021)
	MCM-41	-	-	-	36.1	45.0	248.0	150°C, 8 h	[12] (2019)
	BiOCl/CuNiAl	3.3	5.6	-	11.9	75.4	12.8	60°C, 6 h	[13] (2019)

[1] H. Yan et al. *Angew. Chem.* **2022**, e202116059.

(Reaction conditions: glycerol/glycerol/Pt molar ratio=250, 25 mL glycerol solution(0.05M), 0.5 g NaOH, 50°C, 1 MPa O₂, 16 h)

[2] Zhao M et al. *AICHE*, **2022**, *16*, 34.

(Reaction conditions: 20 mL aqueous solution of GLY (0.105 M), 0.1 g solid catalysts, 60°C, 5 h)

[3] An Z et al. *Nature Comm*, **2022**, *13*, 5467.

(Reaction conditions: 15 mL aqueous solution of GLY (0.10 M), Glycerol/metal=300, 60°C, 8 h)

[4] H. Yan et al. *J Catal.* **370** (2019), 434-449.

(Reaction conditions: 0.2 catalyst, 25 mL aqueous solution of glycerol (0.22 M), glycerol/Pt molar ration 530, 60°C, 1 MPa O₂, 8 h.)

[5] Xu S, Xiao Y, Zhang W, et al. *Chem. Eng. J.* **428** (2022) 132555.

(Reaction conditions: 0.2 g glycerol; 473 K; 0.24 g MgO or 0.16 g MgO; 1 MPa O₂; 400 rpm; 50 mL H₂O)

[7] Yan H, Shen Q, Sun Y, et al. *ACS Catal.* **2021**, *11*, 6371–6383.

(Reaction conditions: 0.05 g catalyst, 25 mL aqueous solution of glycerol (0.105 M), 140 °C, 1MPa O₂, 12h)

[8] X. Jin, M. Zhao, C. Zeng, et al. *ACS Catal.* **6** (2016) 4576-4583.

(Reaction conditions: 0.5 g glycerol, 1.5 g NaOH, 25 mL, 0.2 g solid catalysts. "Others": lactic, glycolic and formic acids, 24 h)

[9] D. Li, H. Gong, et al. *Mol. Catal.* **474** (2019) 110404.

(Reaction conditions: 5 mL glycerol aqueous solution (10 wt.%), 25 mg catalyst, 170 °C, 4 h, 3 MPa O₂)

[10] D. Li, H. Gong, et al. *ACS Sustainable Chem. Eng.* **10** (2022) 17177-17186.

(Reaction conditions: 10 mL glycerol aqueous solution (0.1 M), 20 mg catalyst, 80 °C, 8 h, NaHCO₃/GLY = 4, H₂O₂/GLY=4)

[11] Lin Y, Yang J, Mou C. *ACS Sustainable Chem. Eng.* 2021, 9, 3571–3579

(Reaction conditions: 5 mL of 0.6 mol/L glycerol, glycerol/H₂O₂ molar ratio of 1:5, catalyst amount = 10 mg, 80 °C, 24 h)

[12] Lin Y, Yang J, Mou C. *ACS Sustainable Chem. Eng.* 2021, 9, 3571–3579

(Reaction conditions: 3mL 50%H₂O₂, space velocity of 59.7 h⁻¹, 150 °C, 8 h)

[13] X. Wang, G. Wu, X. Zhang, D. Wang, J. Lan, J. Li, *Catal. Lett.* 149 (2019) 1046-1056.

(Reaction conditions: 25 mL glycerol aqueous solution (0.2 mol/L), 0.2 g catalyst, 60 °C, 3ml 20%H₂O₂, 6 h)

“In order to compare the catalytic performance of MnO₂-D for glycerol oxidation with other reported catalysts, we've categorized the most recently reported catalysts related to this reaction system into three groups: (1) noble metals + O₂ (with other keto acids being the main product), (2) non-noble metals + O₂, and (3) heterogeneous catalyst + H₂O₂.

From the first group (1), we observe that MnO₂-D's catalytic activity in glycerol oxidation using O₂ approaches the levels achieved by precious metals. From the second group (2), it's evident that MnO₂-D delivers the highest catalytic activity and FA selectivity among the non-precious metals using O₂. The third group (3) showcases MnO₂-D's capability to activate O₂, demonstrating performance superior to heterogeneous catalysts utilizing H₂O₂ as an oxidant.”[Supporting Information]

Again, thanks very much for the reviewer's careful review and valuable suggestions.

Reviewer #2:

Yan et.al developed a defective MnO₂ catalyst (MnO₂-D) with intensified Frustrated Lewis Pairs (FLPs) through a coordination number reduction strategy to enhance the aerobic oxidation of various polyols/sugars to formic acid. The tri-coordinated Mn in the defective MnO₂ catalyst exhibits typical characteristics of FLPs, where oxygen vacancies create Mn^{δ+} Lewis acid sites that activate O₂ and nearby electron-reconstructed oxygen creates Lewis base sites to facilitate C–C and C–H bond activation. It is an interesting study. However, there are still some problems that need to be improved before consideration.

Response:

We appreciate the positive comments from reviewer.

1. There are many reports about the solid FLP catalysts. However, the introduction is insufficient to introduce the state-of-the-art progress. Compared with the previous reports, what is the novelty of the method or strategy for constructing the FLP sites.

Response:

We sincerely thank the reviewer for the valuable suggestion.

Based on the suggestion, we have added the current difficulties in the synthesis of solid FLPs in metal oxides, as well as the strategic basis and innovation proposed in this manuscript.

“Due to the tendency of the rigid lattice of traditional metal oxides to form saturated coordination M-O structures, it is still challenging to construct defective metal oxides with FLPs based on existing limited and cumbersome preparation strategies. Inspired by the variable coordination structure of transition metals, rational reduction of the coordination number of the central metal provides a feasible strategy to obtain defective MnO₂ with enhanced FLPs for polyol/sugar cascade oxidation to formic acid⁴⁶⁻⁴⁸.” [Page 3, Line 63-68]

2. After the reaction, characterizations of the spent catalyst using XRD, UV-Vis and XPS techniques prove that the defective Mn^{δ+}-O_v structure in MnO-D was well-preserved during the reaction. Why the MnO₂-D catalysts exhibited the 10% drop during the stability test. Meanwhile, the catalytic activity of MnO₂-D was fully restored. The author is suggested to explain the phenomenon.

Response:

The main reason for the drop of 10% yield is that some carboxylic acid products gradually cover the surface of the catalyst, resulting in the deactivation of some sites. After high-temperature re-calcination, the carboxylic acid species covered on the surface were removed, thereby restoring catalytic performance. To prove this point, we conducted IR characterization of MnO₂-D before and after the reaction. Fig. R6 shows that both spent MnO₂-P and MnO₂-D exhibit stronger peaks of stretching vibration of C=O and O-H bonds and deformation vibration of C-O bond compared to fresh catalysts, indicating that the ketone acids (and their condensation products) could partially cover the surface of the catalyst after the reaction, which is also confirmed by Qu et. al. (ACS Catal. 2020, 10, 3832–3837). This resulted in a partial decrease in the FA yield of MnO₂-D. After the re-calcination of spent catalyst, the carboxylic

acid species covered on the MnO₂-D surface could be removed. Detailed explanations have been added into the revised manuscript.

Figure R6. IR spectra of the spent MnO₂-P, spent MnO₂-D, fresh MnO₂-P and spent MnO₂-D.

“Moreover, MnO₂-D exhibits excellent stability for glycerol oxidation, with almost no change in FA selectivity (>80%) and a slight decrease in glycerol conversion (<20%) after 6 catalytic cycles due to the coverage of some surface active sites by carboxylic acid products (Fig. 3c and Supplementary Fig. 5). Characterizations of the spent catalyst using XRD, UV-Vis, XPS, NH₃-TPD and CO₂-TPD techniques prove that the defective Mn^{δ+}-O_v structure and acid-base properties in MnO₂-D were well-preserved during the reaction (Supplementary Fig. 5).”
[Page 10, Line 203-208]

“The main reason for the drop of 10% yield is that some carboxylic acid products gradually cover the surface of the catalyst, resulting in the deactivation of some sites. To prove this point, we conducted IR characterization of MnO₂-D before and after the reaction. Supplementary Fig. 5g shows that both spent MnO₂-P and MnO₂-D exhibit stronger peaks of stretching vibration of C=O and O-H bonds and deformation vibration of C-O bond compared to fresh catalysts, indicating that the ketone acids (and their condensation products) could partially cover the surface of the catalyst after the reaction. This resulted in a partial decrease in the FA yield of MnO₂-D. After the re-calcination of spent catalyst, the carboxylic acid species covered on the MnO₂-D surface could be removed, thus restoring the catalytic performance.”[Supporting Information]

3. The TOF value. The active sites on MnO₂-D were determined by ethanol adsorption. How does this TOF value compare to other reported results?

Response:

We've incorporated additional catalyst reports from the literature for comparison with MnO₂-D. As demonstrated in Supplementary Table 7, the TOF value of MnO₂-D approaches the level of noble metal catalysts for glycerol oxidation using O₂, while significantly surpassing that of other non-noble metal catalysts. We've integrated these relevant details into the updated manuscript.

Supplementary Table 7. Oxidation of glycerol over MnO₂-D catalysts and reported catalysts

	Catalyst	Selectivity (%)				Conversion (%)	TOF (h ⁻¹)	Reaction conditions	Reference (Year)
		GLYA	GLYOA	TA	FA				
MnO₂-D+O₂	MnO ₂ -D	1.4	3.2	0.0	83.2	99.2	113.5	120°C, 1MPa, 8 h	This work
Noble metals + O₂	Pt ₁ /HAP	90.3	1.8	7.8	0.0	91.2	351.2	50°C, 1MPa, 16 h	[1] (2022)
	AuCu/ZnO	3.6	1.1	-	-	3.1	75.4	60°C, 1MPa, 8 h	[2] (2022)
	Pt ₁ +Pt _n /Cu-CuZrO _x	80.0	1.1	1.4	0.0	90.1	224.0	60°C, 1bar, 8 h	[3] (2022)
	Pt/MgO/SAB-15 (0.1)	65.5	5.4	2.0	0.9	60.2	1671.2	60°C, 1MPa, 8 h	[4] (2019)
	Pt/SAB-15	49.9	5.0	2.0	0.1	25.5	121.0	60°C, 1MPa, 8 h	[4] (2019)
Non noble metals + O₂	Cu ₁ Mg ₄ O _x	8.9	60.3	0.1	22.0	89.2	0.2	200°C, 1MPa, 4 h	[5] (2022)
	Mn ₂ O ₃ -T	9.8	41.7	-	-	68.6	0.4	140°C, 1MPa, 12h	[6] (2021)
	Ni ₁ Co ₁ O _x	66.5	19.5	-	6.9	78.5	5.4	80°C, 1MPa, 24 h	[7] (2020)
	Co _{0.15} /Mg ₃ Al-c	29.2	-	46.9	13.4	1.1	0.7	70°C, 1MPa, 24 h	[8] (2016)
	Co _{0.15} /Mg ₃ Al-s	2.1	-	63.5	2.4	100	2.4	70°C, 1MPa, 24 h	[8] (2016)
	V ₃ /ZrP-m	-	-	-	50.9	89.0	0.7	170°C, 3MPa, 4 h	[9] (2019)
	CuN ₄ /NCSWCs	0.0	6.4	0.0	61.3	96.3	1.1	80°C, 8 h	[10] (2022)
Heterogeneous catalyst +H₂O₂	Au-PTA/MSN	-	-	-	79.2	33.6	7.1	80°C, 24 h	[11] (2021)
	MCM-41	-	-	-	36.1	45.0	248.0	150°C, 8 h	[12] (2019)
	BiOCl/CuNiAl	3.3	5.6	-	11.9	75.4	12.8	60°C, 6 h	[13] (2019)

[1] H. Yan et al. *Angew. Chem.* 2022, e202116059.

(Reaction conditions: glycerol/glycerol/Pt molar ratio=250, 25 mL glycerol solution(0.05M), 0.5 g NaOH, 50°C, 1 MPa O₂, 16 h)

[2] Zhao M et al. *AICHE*, 2022, 16, 34.

(Reaction conditions: 20 mL aqueous solution of GLY (0.105 M), 0.1 g solid catalysts, 60°C, 5 h)

[3] An Z et al. *Nature Comm*, 2022, 13, 5467.

(Reaction conditions: 15 mL aqueous solution of GLY (0.10 M), Glycerol/metal=300, 60°C, 8 h)

[4] H. Yan et al. *J Catal.* **370** (2019), 434-449.

(Reaction conditions: 0.2 catalyst, 25 mL aqueous solution of glycerol (0.22 M), glycerol/Pt molar ration 530, 60°C, 1 MPa O₂, 8 h.)

[5] Xu S, Xiao Y, Zhang W, et al. *Chem. Eng. J.* **428** (2022) 132555.

(Reaction conditions: 0.2 g glycerol; 473 K; 0.24 g MgO or 0.16 g MgO; 1 MPa O₂; 400 rpm; 50 mL H₂O)

[7] Yan H, Shen Q, Sun Y, et al. *ACS Catal.* **2021**, *11*, 6371–6383.

(Reaction conditions: 0.05 g catalyst, 25 mL aqueous solution of glycerol (0.105 M), 140 °C, 1MPa O₂, 12h)

[8] X. Jin, M. Zhao, C. Zeng, et al. *ACS Catal.* **6** (2016) 4576-4583.

(Reaction conditions: 0.5 g glycerol, 1.5 g NaOH, 25 mL, 0.2 g solid catalysts. "Others": lactic, glycolic and formic acids, 24 h)

[9] D. Li, H. Gong, et al. *Mol. Catal.* **474** (2019) 110404.

(Reaction conditions: 5 mL glycerol aqueous solution (10 wt.%), 25 mg catalyst, 170 °C, 4 h, 3 MPa O₂)

[10] D. Li, H. Gong, et al. *ACS Sustainable Chem. Eng.* **10** (2022) 17177-17186.

(Reaction conditions: 10 mL glycerol aqueous solution (0.1 M), 20 mg catalyst, 80 °C, 8 h, NaHCO₃/GLY = 4, H₂O₂/GLY=4)

[11] Lin Y, Yang J, Mou C. *ACS Sustainable Chem. Eng.* **2021**, *9*, 3571–3579

(Reaction conditions: 5 mL of 0.6 mol/L glycerol, glycerol/H₂O₂ molar ratio of 1:5, catalyst amount = 10 mg, 80 °C, 24 h)

[12] Lin Y, Yang J, Mou C. *ACS Sustainable Chem. Eng.* **2021**, *9*, 3571–3579

(Reaction conditions: 3mL 50%H₂O₂, space velocity of 59.7 h⁻¹, 150 °C, 8 h)

[13] X. Wang, G. Wu, X. Zhang, D. Wang, J. Lan, J. Li, *Catal. Lett.* **149** (2019) 1046-1056.

(Reaction conditions: 25 mL glycerol aqueous solution (0.2 mol/L), 0.2 g catalyst, 60 °C, 3ml 20%H₂O₂, 6 h)

4. The EPR results. In Figure 4b, the quantitative relation is suggested to further clarified.

Response:

Taking the comment into consideration, we have added analysis and discussion on in situ EPR. Fig. R7a shows that the symmetrical EPR peaks at $g=2.003$ ascribed to unpaired electrons in the oxygen vacancies on MnO₂-D. The concentration of oxygen vacancies is reflected by the signal intensity (ACS Appl. Mater. Interfaces 2022, 14, 18525–18538). As the reaction temperature increases, the signal gradually increases, indicating the enhanced activation of unpaired electrons on oxygen vacancies. This observation encouraged us to explore a potential correlation between the initial reaction rate and oxygen vacancy signal intensity, yielding a strong linear relationship. Consequently, we infer that an increase in the oxygen vacancy signal signifies a proportional rise in catalytic activity. We've incorporated these discussions into the updated manuscript.

Figure R7. (a) *In situ* EPR spectra for the detection of the evolution of oxygen vacancies on MnO₂-D.

(b) The linear relationship between initial reaction rate (mol/L/h) and EPR intensity.

“We observe a symmetrical EPR peak at $g=2.003$ in MnO₂-D, attributable to unpaired electrons associated with oxygen vacancies of metal oxides. Interestingly, the intensity of the peak shows a progressive increase with rising reaction temperature. Moreover, the peak intensity demonstrates a strong linear correlation with initial catalytic activity, indicating that an increase in the oxygen vacancy signal corresponds to a proportional increase in catalytic activity.” [Page 11-12, Line 227-231]

Reviewer #3:

The manuscript from Yan et al explores the use of defective manganese oxide to catalyse the oxidation of a range of polyols to formic acid. The MnO₂ with more oxygen vacancies (MnO₂-D) has an enhanced catalytic performance compared to the nominally pristine MnO₂ sample (MnO₂-P). This discovery is in line with a large number of reports on the role of defects in promoting heterogeneous catalysis (see recent review for many examples: <https://pubs.acs.org/doi/pdf/10.1021/acscatal.0c03034>).

The article is well written and clearly laid out. The materials appear to be well characterised, and I believe enough experimental details are provided for other researchers to replicate these results. The evidence clearly shows the MnO₂-D is a more efficient catalyst than MnO₂-P (Figure 3a). The recycling experiments and conversion of postulated intermediates to formic acid provides nice mechanistic information, and the control experiments have been well done; for example, the data in Supplementary Figure 5 to rule out the importance of specific surface area between the two catalysts.

Response:

We appreciate the positive comments from the reviewer. The related example articles have been included in the revised manuscript as well.

1. I do question the validity of invoking the concept of frustrated Lewis pairs (FLPs) to describe this work and mechanism. There are examples in the literature of using FLPs to describe the

reactivity at surfaces in heterogeneous catalysis; some of these the authors have cited, but there are key references missing for this topic:

- FLPs on the surface of alumina (<https://onlinelibrary.wiley.com/doi/10.1002/anie.201006794> and <https://pubs.acs.org/doi/10.1021/ja3042383>) these articles are the first that relate reactivity at vacancies to FLPs

- There has also been extensive work on indium oxide surfaces, typified by this article: <https://pubs.rsc.org/en/content/articlelanding/2015/CP/C5CP02613J>

Response:

Thanks for the comment. Relevant works related to FLPs on alumina and indium oxide have been added into the revised manuscript. These works (Angew. Chem. Int. Ed. 2011, 50, 3202–3205; J. Am. Chem. Soc. 2012, 134, 14430–14449; Phys. Chem. Chem. Phys. 2015, 17, 14623; ACS Cent. Sci. 2015, 1, 313–319; J. Am. Chem. Soc. 2015, 137, 10018–10032) demonstrated the usefulness of FLPs on heterogeneous oxide catalysts.

“These co-existing acidic and basic sites in metal oxides could be regarded as frustrated Lewis pairs (FLPs), which promote reactions on the catalyst surfaces that require both acid and base functions, as previously demonstrated on (hydroxylate-)Al₂O₃, InO_x, CeO_x, MoO_x and CoO_x oxides²⁸⁻³⁶.” [Page 3, Line 53-57]

2. However, in these examples the surfaces do actually behave somewhat like FLPs; i.e. you have a Lewis acidic site and Lewis basic site that can carry out cooperative reactivity on a small molecule. In this current manuscript from Yan et al, there are acidic sites (Mn) and basic sites (O), and the authors show very nicely that the relative acidity and basicity of these sites is enhanced in MnO₂-D (Figure 2g-1 and 2-g2). However, the reaction that is then catalysed (oxidation of glycerol) bears little resemblance to a reaction that goes via Lewis acidic and basic sites; instead the first step in Figure 4c shows two adjacent Mn centres activating O₂ and water to give a hydroxyl radical and the “Lewis base” is not involved at all. The ensuing cascade mainly involves the generation and transfer of hydroxyl radicals to glycerol, as stated at the bottom of page 11: “O₂ mainly reacts with H₂O to generate hydroxyl radical to participate in the subsequent activation of glycerol, which is consistent with previous reports [ref 47]” While

there is growing interest in the use of radicals in FLP chemistry, this proposed mechanism does not correlate to any of those, as the reactivity is just at the Mn centres (which are more reactive in MnO₂-D due to being low-coordinate relative to MnO₂-P).

Response:

We sincerely thank the reviewer for the insightful comment.

From the perspective of O₂ activation, its activation is mainly promoted by Mn acidic sites, but also requires indirect assistance of O_V basic sites. The activation paths of the two O atoms in O₂ are different. One O is directly anchored by the Mn center, while the second O is activated by H dissociated from H₂O. This is somewhat similar to the synergistic effect between the hydroxyl and Al sites on the surface of γ -Al₂O₃ (J. Am. Chem. Soc. 2012, 134, 14430-14449). From the perspective of H₂O dissociation, H₂O is dissociated into OH adsorbed on the Mn center and H adsorbed on adjacent O_{Mn}. Notably, adjacent O_{Mn} could come from both adsorbed O₂ and defective Mn–O structures. This process is better aligned with the definition of metal Lewis acid-base sites, where one accepts OH and the other accepts H.

Fig. 4: Oxygen activation into hydroxyl radical on Lewis acid site in MnO₂-D. a In situ EPR spectra for the detection of the evolution of oxygen vacancies on MnO₂-D. **b** In situ EPR spectra with free radical trapping agent (DMPO) for the oxidation of glycerol on MnO₂-P and MnO₂-D (for the capitiation of hydroxyl radical). **c** Electron transition in orbit during O₂ adsorption over Mn^{δ+}-O_V structure.

Further considering this valuable comment, the definition of FLPs in metal oxides (Al_2O_3 , In_2O_3 , CeO_2 , etc.) are consistent with that in MnO_2 —these special combinations of donor and acceptor, sterically frustrated from the formation of classic adducts. For the activation of H_2 and CH_4 , this process can be intuitively achieved in one step on L acid (metal cation) and L base (nearby O or OH). H_2 is hetero-cleaved into $\text{M}-\text{H}^{\delta-}$ and $(\text{H})\text{O}-\text{H}^{\delta+}$ on the $\text{M}-\text{O}(\text{H})$ structure, while CH_4 is activated into $\text{M}-\text{CH}_3^{\delta-}$ and $(\text{H})\text{O}-\text{H}^{\delta+}$. If compared to this process, the dissociation of H_2O assisted by O_2 (including the subsequent activation of O-H on hydroxyl groups of alcohols discussed later) is somewhat similar to it. As the reviewer mentioned, polyol oxidation is a complex system that is difficult to fully reflect the role of FLPs. We therefore analyzed the two most critical steps in this process (O_2 dissociation and C-H bond activation). The reason why it differs from typical FLPs processes is that this is an oxidation reaction, and the reactant contains multiple O atoms. The O in these substrates tends to bind with Mn metal sites, which in turn brings synergistic catalytic effects. Specifically, the Mn metal site, along with the bound O, acts as a site to synergistically catalyze with the O_{metal} in its own Mn-O structure, achieving the activation of O_2 and C-H bonds. Relevant discussions have been added into the revised manuscript.

“The activation of O_2 is mainly promoted by $\text{Mn}^{\delta+}$ acid site with the assistant of O_V basic sites since one O atom is directly anchored by the Mn center, while another O atom is activated by H dissociated from H_2O . From the perspective of H_2O dissociation, H_2O is dissociated into OH adsorbed on the Mn center and H adsorbed on adjacent O_{Mn} or O_V . The latter process better resembles the traditional narrow definition of Lewis acid-base, where one accepts OH and the other accepts H. Finally, the $\text{Mn}^{\delta+}$ Lewis acid could easily accept the as-formed OH to form $\text{Mn}^{\delta+}-\text{OH}$.”[Page 12, Line 242-248]

3. The dehydrogenation steps are the closest to FLP reactivity (e.g. from R1 to P3, where H_2 is lost from the starting glycerol, or R7 to P8), but even here the first H is removed as H-radical by a hydroxyl radical (not by either component of the “FLP”), and then there is a second homolytic cleavage where another H-radical is transferred to an oxygen in the catalyst, so this isn't typical FLP reactivity either.

Response:

Thanks for the insightful suggestion from the reviewer.

Prompted by this comment, we revisited and optimized the reaction pathway calculated by DFT, specifically re-evaluating the dehydrogenation steps (O–H and C–H bonds) using R1 to P2 as examples. On revisiting the activation of the O–H bond in the first step, we discovered that the $\text{Mn}^{\delta+}\text{-O}_V$ structure can accomplish this directly without the need for a hydroxyl radical. As depicted in Fig. R8, the glycerol's primary hydroxyl group's oxygen atom is bound by $\text{Mn}^{\delta+}$ (acting as a Lewis acid), while the hydrogen atom is abstracted by the adjacent O_V (acting as a Lewis base), thereby achieving dehydrogenation of the O–H bond. This mechanism aligns better with traditional understanding of FLPs catalyzed small molecule activation. Notably, the activation energy of this direct dehydrogenation via the $\text{Mn}^{\delta+}\text{-O}_V$ structure is lower than that of previously considered free radicals. We thank the reviewer's comment which helped us to identify an energetically more favored path for O–H activation.

Moreover, in the dehydrogenation of the C–H bond, while it appears that the hydrogen atom's attraction is primarily facilitated by the O_V acting as a Lewis base, the Mulliken charge analysis indicates a significant role of the $\text{Mn}^{\delta+}$ acid sites. Specifically, the hydrogen atom moving from the C–H bond to the $\text{Mn}^{\delta+}\text{-O}_V$ structure consistently retains a partial positive charge ($\text{O}_V\text{-H}^{\delta+}$), while the carbon atom in the C–H bond, influenced by the Mn metal site, initially exhibits a partial negative charge ($\text{Mn}^{\delta+}\text{-substrate}^{\delta-}$). This charge distribution is analogous to the charge transfer process in FLPs activating H_2 or CH_4 . The specific oxygen enrichment in the oxidation system likely enables the Mn site to form an acidic site by combining with oxygen-bearing substrates, thus working cooperatively with the electron-reconstructed oxygen basic site to activate the C–H bond. We've integrated these discussions into the updated manuscript.

Previous Step-1: $\text{RCH}_2\text{OH}^* + \text{OH}^* \rightarrow \text{H}_2\text{O}^* + \text{RCH}_2\text{O}^*$ Promoted by hydroxyl radical **Dehydrogenation of O-H bond**

Now Step-1: $\text{RCH}_2\text{OH}^* + \text{H}^* \rightarrow \text{RCH}_2\text{O}^* + \text{H}_2\text{O}^*$ ($\text{H}^* + \text{OH}^* \rightarrow \text{H}_2\text{O}^*$) Promoted by $\text{Mn}^{\delta+}\text{-O}_v$ structure

Step-2: $\text{RCH}_2\text{O}^* + \text{H}^* \rightarrow \text{RCHO}^* + \text{H}^*$

Dehydrogenation of C-H bond

Figure R8. Mulliken charge ($|e|$) distribution of the five coordinated Mn and three coordinated Mn in the $\text{MnO}_2\text{-P}$ (Penta) and $\text{MnO}_2\text{-D}$ (Tri). Configuration diagrams of reactants, transition states, and products on the $\text{MnO}_2\text{-P}$ (Penta) [blue] and $\text{MnO}_2\text{-D}$ (Tri) [red] (Step-1 to Step-2)

"Fig. 5b presents a schematic of the oxidation mechanism of glycerol to FA via $\text{MnO}_2\text{-D}$. Initially, the $\text{Mn}^{\delta+}\text{-O}_v$ structure utilizes $\text{Mn}^{\delta+}$ to adsorb oxygen from the O-H bond, while the O_v attracts hydrogen from the O-H bond, thereby achieving the dehydrogenation of the O-H bond. In the subsequent step, the synergistic adsorption of oxygen-containing intermediates at the $\text{Mn}^{\delta+}$ metal site assists O_v in attracting hydrogen from the C-H bond, leading to the formation of glyceraldehyde." [Page 15, Line 283-287]

4. It must be said that the proposed mechanism is quite challenging to follow and unnecessarily time-consuming to understand properly. Figure 5 in the manuscript should contain a schematic of the mechanism instead of the strings of text underneath the energy diagrams in section b. Furthermore, there are a host of errors or confusing details in the configuration diagrams in Supplementary Figure 11:

- (A) R1 and P1 in MnO₂-D are the same image, clearly R1 is wrong
- (B) R2 and P2 in MnO₂-D are the wrong way round
- (C) P(n) and R(n+1) should be the same for the energy diagram in Figure 5 to make any sense. For MnO₂-D, P1 has the central OH bond activated, but then in R2 (notwithstanding R2 and P2 are the wrong way round), it is one of the terminal OH groups that is activated, and it must be the latter case for the subsequent reactivity so P1 is presumably wrong?
- (D) For MnO₂-P, R4 is wrong and there has been migration of an H atom from C to O (compared to P3 and the formula above the panel, and compared to the same species in MnO₂-D. TS4 and P4 are also mislabelled as TS10 and P10
- (E) Furthermore, R4 in MnO₂-P is also missing the hydroxyl group that is required for the attack
- (F) For MnO₂-D, there is a large rearrangement between R4 and TS4. Can the authors please confirm that IRC calculations have been carried out on transition states to show that the R(n) and P(n) are actually linked by TS(n) in each case?
- (G) The caption to Supplementary Figure 11-3 states that there is a “Mulliken charge distribution diagram of the rate-determining step (Step 10) in the MnO₂-P and MnO₂-D” - this appears to be missing (and there is no step 10 at all?)

Response:

Thanks for the reviewer’s careful check of the figure and data.

Based on the suggestion, we have changed Fig. 5b to a schematic diagram to provide a clearer representation of the glycerol oxidation to FA reaction mechanism over MnO₂-D. As the figure illustrates, initially, the Mn^{δ+} site in the Mn^{δ+}-O_v structure adsorbs O in the O–H bond, while O_v captures hydrogen in the O–H bond to achieve dehydrogenation of the hydroxyl group. Subsequently, assisted by the synergistic adsorption of oxygen-containing intermediates at Mn^{δ+} metal site, O_v further captures hydrogen in the C–H bond to form glyceraldehyde. Then, under the attack of hydroxyl radical, the newly formed glyceraldehyde rapidly transforms into CH₂OHCHOHCHOOH, which undergoes C–C bond cleavage over the Mn^{δ+}-O_v structure. After a series of processes comprising multiple O–H bond dehydrogenations, C–H bond dehydrogenation, free radical attack, and C–C bond cleavage processes mentioned above, the original glycerol is converted into three molecules of FA product. In this reaction, hydroxyl

radical primarily serves two functions: firstly, it attacks the aldehyde intermediate, and secondly, it combines with the hydrogen removed from the O–H and C–H bonds to produce water, thus regenerating the $\text{Mn}^{\delta+}\text{-O}_V$ active site.

Fig. 5: Mechanistic investigations for cascade oxidation of polyol/sugar to formic acid. a In situ FT-IR spectra of glycerol oxidation to formic acid over $\text{MnO}_2\text{-P}$ and $\text{MnO}_2\text{-D}$. **b** Schematic diagram of reaction mechanism for the oxidation of glycerol to formic acid. **c** Partial density of states (PDOS) of $\text{MnO}_2\text{-P}$ (Penta) and $\text{MnO}_2\text{-D}$ (Tri). **d** Activity map for glycerol oxidation.

To thoroughly rectify the inaccuracies and potentially confusing details in the configuration diagrams presented in Supplementary Figure 12, we have conducted calculation based on the modified reaction pathway and updated the configuration diagrams of the substrate, transition state, and product. The modifications primarily relate to two elementary steps: 1) The O–H bond dehydrogenation, which we revised from being promoted by a hydroxyl group to being facilitated by the $\text{Mn}^{\delta+}\text{-O}_V$ defect structure. 2) The C–C bond cleavage, which we

adjusted from a single step (directly attacked by OH) to a two-step process (initial C–C bond cleavage followed by OH attack).

The specific modifications are as follows:

(A) The dehydrogenation of the O–H bond has been recalculated, and the corresponding configurations have also been corrected.

(B) R2 and P2 have been corrected.

(C) Previously, there was an error in depicting the secondary hydroxyl activation process in MnO₂-D, which has now been rectified.

(D) (E) (F) The elemental steps of C–C bond cleavage have been revised. Intermediate CH₂OHCHOHCHOOH* first undergoes cleavage of C–C bond directly to expose the C atom, followed by an attack from a hydroxyl radical to form a subsequent reaction intermediate CH₂OHCHOHOH*. The above process has been described in a two-step elementary reaction: CH₂OHCHOHCHOOH*+* → CH₂OHCHOH*+CHOOH* and CH₂OHCHOH*+OH* → CH₂OHCHOHOH*.

(G) The caption to Supplementary Figure 12 has been updated.

“Fig. 5b presents a schematic of the oxidation mechanism of glycerol to FA via MnO₂-D. Initially, the Mn^{δ+}–O_V structure utilizes Mn^{δ+} to adsorb oxygen from the O–H bond, while the O_V attracts hydrogen from the O–H bond, thereby achieving the dehydrogenation of the O–H bond. In the subsequent step, the synergistic adsorption of oxygen-containing intermediates at the Mn^{δ+} metal site assists O_V in attracting hydrogen from the C–H bond, leading to the formation of glyceraldehyde. Under the attack of hydroxyl radical, the as-formed glyceraldehyde is rapidly transformed into CH₂OHCHOHCHOOH, which undergoes C–C bond cleavage over Mn^{δ+}–O_V structure. After a series of processes comprising multiple O–H bond dehydrogenations, C–H bond dehydrogenation, free radical attack, and C–C bond cleavage steps mentioned above, the original glycerol is converted into three molecules of FA product.”[Page 15, Line 283-291]

Supplementary Fig. 12-1 Free energy diagrams for the oxidation of glycerol to formic acid on the $\text{MnO}_2\text{-P}$ (Penta) [blue] and $\text{MnO}_2\text{-D}$ (Tri) [red]

Step-1: $\text{CH}_2\text{OHCHOHCH}_2\text{OH}^+ +^* \rightarrow \text{CH}_2\text{OHCHOHCH}_2\text{O}^* + \text{H}^+$ ($\text{OH}^* + \text{H}^* \rightarrow \text{H}_2\text{O}^*$) **Dehydrogenation of O-H bond**

Step-2: $\text{CH}_2\text{OHCHOHCH}_2\text{O}^* +^* \rightarrow \text{CH}_2\text{OHCHOHCHO}^* + \text{H}^+$ ($\text{OH}^* + \text{H}^* \rightarrow \text{H}_2\text{O}^*$) **Dehydrogenation of C-H bond**

Step-3: $\text{CH}_2\text{OHCHOHCHO} + \text{OH}^* \rightarrow \text{CH}_2\text{OHCHOHCHOOH}^* +^*$ **Oxidation of aldehyde**

Step-4: $\text{CH}_2\text{OHCHOHCHOOH}^* +^* \rightarrow \text{CH}_2\text{OHCHOH}^* + \text{CHOOH}^*$ **C-C bond cleavage**

Supplementary Fig. 12-2 Configuration diagrams of reactants, transition states, and products on the MnO₂-P (Penta) [blue] and MnO₂-D (Tri) [red] (Step-1 to Step-4)

Oxidation of aldehyde

Dehydrogenation of O-H bond

C-C bond cleavage

Oxidation of aldehyde

Supplementary Fig. 12-3 Configuration diagrams of reactants, transition states, and products on the MnO₂-P (Penta) [blue] and MnO₂-D (Tri) [red] (Step-5 to Step-8)

Step-9: $\text{CH}_2\text{OH}^+ + \cdot \rightarrow \text{CH}_2\text{OOH}^+ + \text{H}^+$ ($\text{OH}^+ + \text{H}^+ \rightarrow \text{H}_2\text{O}^+$) Dehydrogenation of O-H bond

Step-10: $\text{CH}_2\text{OOH}^+ + \cdot \rightarrow \text{CHOOH}^+ + \text{H}^+$ ($\text{OH}^+ + \text{H}^+ \rightarrow \text{H}_2\text{O}^+$) Dehydrogenation of C-H bond

Supplementary Fig. 12-4 Configuration diagrams of reactants, transition states, and products on the $\text{MnO}_2\text{-P}$ (Penta) [blue] and $\text{MnO}_2\text{-D}$ (Tri) [red] (Step-9 to Step-10). Mulliken charge ($|e|$) distribution of the rate-determining step (Step-10) in the $\text{MnO}_2\text{-P}$ (Penta) and $\text{MnO}_2\text{-D}$ (Tri).

5. Overall, I think the experimental part of the paper is very good and demonstrates that $\text{MnO}_2\text{-D}$ is a good catalyst for the oxidation of glycerol, but the DFT mechanistic investigation needs to be sorted out before the paper can be published. Furthermore, I don't see what the authors have to gain by labelling this as "FLP" chemistry; it is a radical cascade mechanism that is promoted very well by the MnO_2 system (and outcompetes precious metal systems according to Figure 3b). That is not to say that this system couldn't behave as an FLP under different reaction conditions, as the authors have shown that there are accessible acidic and basic sites on the surface, but this reaction is not a good example to demonstrate this point.

Response:

Prompted by the comment, we have newly conducted DFT calculation to better elucidate the potential involvement of FLPs in O₂ and H₂O co-activation, O–H bond activation and C–H bond activation. For O₂ and H₂O co-activation, H₂O is dissociated into OH adsorbed on the Mn center and H adsorbed on adjacent O_{Mn}. The activation of O₂ is mainly promoted by Mn^{δ+} acid site with the assistant of O_V basic sites since one O atom is directly anchored by the Mn center, while another O atom is activated by H dissociated from H₂O. For O–H bond activation, the glycerol's primary hydroxyl group's oxygen atom is bound by Mn^{δ+}, while the hydrogen atom is abstracted by the adjacent O_V, thereby achieving dehydrogenation of the O–H bond. This re-calculated dehydrogenation of O–H bond via the Mn^{δ+}–O_V structure is energetically more favored than that of previously considered free radicals. For C–H bond activation, the hydrogen atom's attraction is primarily facilitated by the O_V, and the carbon atom in the C–H bond, influenced by the Mn metal site, works cooperatively with the electron-reconstructed oxygen basic site to activate the C–H bond. Additionally, we have also re-calculated the C–C bond cleavage, identifying an energetically more favored route for a two-step process (initial C–C bond cleavage followed by OH attack) against originally proposed a single step (directly attacked by OH). By these efforts, we hope we have enhanced the DFT part in the study.

Nonetheless, we fully agree with the reviewer and acknowledge that our system may not be the most commonly seen scenario for FLPs promoted reactions. Therefore, we have revised the title to “*Enhancing Polyol/Sugar Cascade Oxidation to Formic Acid with Low-Coordination Mn^{δ+}-O_V Structure*” to tune down the claim on FLPs. The logic flow in the introduction has also been revised accordingly. We still retain the discussion on the prospective roles of FLPs in the reaction based on both experimental and theoretical evidences.

REVIEWER COMMENTS

Reviewer #1 (Remarks to the Author):

I would like to thank the authors for their careful consideration of my comments and their detailed revision of the manuscript. My main concerns have now been addressed and I now consider there is sufficient evidence to justify the authors primary conclusions. I have just a few minor comments remaining, which I would like the authors to address.

1. The calculation the authors use to calculate the carbon mass balance seems unusual. Why do they need to multiply the moles of glycerol by three? I would recommend they recalculate the carbon mass balances using a more traditional method.

i.e. $CMB (\%) = (\text{moles of carbon from products}) / (\text{moles of carbon from substrate}) \times 100$

e.g. if only 1 % of glycerol is converted to ca. 0.7 % of products the CMB should be 99.7 %, shouldn't it? 0.3 % of the carbon in is missing.

2. I appreciate the authors attempt to use radical scavengers to confirm the active species in the reaction. However, using organic radical scavengers might influence the catalytic activity in other ways (e.g. through competitive adsorption). It may have been more useful to use an inorganic scavenger for these experiments. Furthermore, the references the authors have used to support the formation of hydroxyl/peroxy radicals in their response document don't seem (to me at least) to be relevant. In Davis and co-workers paper, the authors don't suggest that oxygen reduction by water results in the homolytic cleavage of O₂. This is what what would be require to produce hydroxyl radicals, wouldn't it? The authors state in their revised text that hydroxyl radicals are produced through the interaction of O₂ with H₂... is this correct?

Reviewer #2 (Remarks to the Author):

I am satisfied with the authors' response. It could be accepted for the revised manuscript.

Reviewer #3 (Remarks to the Author):

I am satisfied that the authors have addressed my concerns from the original peer review, particularly regarding the mechanism. The concerns of the other reviewers also appear to have been addressed well. The authors have also extended the study at the request of one of the other reviewers, and overall I think the manuscript is stronger for it. Overall I believe the article is now suitable for acceptance in Nat Comm, and the authors should be commended for doing such a good and thorough job with their revisions.

Responses to Reviewers' Comments

Reviewer #1:

I would like to thank the authors for their careful consideration of my comments and their detailed revision of the manuscript. My main concerns have now been addressed and I now consider there is sufficient evidence to justify the author's primary conclusions. I have just a few minor comments remaining, which I would like the authors to address.

1. The calculation the authors use to calculate the carbon mass balance seems unusual. Why do they need to multiply the moles of glycerol by three? I would recommend they recalculate the carbon mass balances using a more traditional method.

i.e. $\text{CMB (\%)} = (\text{moles of carbon from products})/(\text{moles of carbon from substrate}) \times 100$

e.g. if only 1 % of glycerol is converted to ca. 0.7 % of products the CMB should be 99.7 %, shouldn't it? 0.3 % of the carbon in is missing.

Response:

Thanks for the reviewer's comment.

Based on your suggestion, we recalculate the carbon mass balances using a more traditional method. The definition of carbon mass balance (CMB) was calculated by the following formula:

$$\text{CMB} = \frac{\sum_{i=1}^3 i \times \text{moles of } C_i \text{ product (including unconverted glycerol)}}{3 \times \text{moles of glycerol substrate}} \times 100\%$$

All the data of CMB in the Supporting Information have been recalculated and updated. The reason why the mole number of glycerol is multiplied by three times is that the oxidation of glycerol with three carbon numbers may generate oxidation products with different carbon numbers including C3 (glyceric acid, etc.), C2 (oxalic acid and Glycolic acid) and C1 products (formic acid and CO₂). To facilitate the statistics of carbon moles, we normalized both reactants and products to obtain the carbon balance based on the carbon number. We have integrated these results into the updated manuscript.

Supplementary Table 6-1. Oxidation of glycerol and other substrates over Mn-based catalysts

Catalyst	Selectivity (%)					Conversion (%)	TOF (h ⁻¹)	Initial reaction rate (mmol/h/g _{cat})	C%
	GLYA	GLYOA	OA	FA	Others				
Blank (no catalyst)	26.7	28.5	14.5	0.0	2.1	1.0	-	-	99.7
MnO₂-P^a	7.1	14.5	21.8	23.8	31.4	63.8	9.6	1.6	99.1
MnO₂-D^b	4.3	4.1	3.6	83.3	3.6	63.9	113.5	31.3	99.3
MnO^c	44.4	3.6	0.0	0.0	48.9	5.9	-	-	99.8
Mn₂O₃^c	39.7	1.6	0.0	1.1	52.8	5.1	-	9	99.8
MnO₂^c	70.2	3.5	0.0	0.2	25.0	5.1	-	-	99.9
Ethylene glycol (EG)^d	0.1	2.5	0.0	84.5	11.1	99.1	-	-	98.2
1,2-propanediol (PG-2)^e	0.0	0.0	0.0	66.5	31.5	77.4	-	-	98.5
1,3-propanediol (PG-3)^e	0.0	0.0	0.0	65.7	30.5	82.5	-	-	96.9
Erythritol (ET)^f	0.3	0.0	3.5	95.0	0.4	99.8	-	-	99.2
Xylitol (XT)^f	0.0	0.0	3.9	87.6	5.8	99.7	-	-	97.3
Sorbitol (ST)^f	0.0	1.6	8.0	82.5	4.4	99.8	-	-	96.5
Formic acid (FA)^e	0.0	0.0	0.0	-	81.5	5.5	-	-	99.0
Glycolic acid (GLYOA)^e	0.0	-	1.5	91.5	4.0	99.1	-	-	97.0
Oxalic acid (OA)^g	0.0	0.0	-	85.1	10.2	94.3	-	-	95.6
Glycerol (GLYA)^e	-	10.2	2.2	80.5	2.8	99.6	-	-	95.7
Tartronic acid (TA)^g	0.0	10.5	6.1	78.2	3.6	99.1	-	-	98.4

“The definitions of conversion (X), product selectivity (S), turnover frequency (TOF) and carbon mass balance (CMB) were calculated by the following formula: ”[Page 17, Line 336-342]

$$\text{CMB} = \frac{\sum_{i=1}^3 i \times \text{moles of } Ci \text{ product (including unconverted glycerol)}}{3 \times \text{moles of glycerol substrate}} \times 100\%$$

2. A) I appreciate the authors attempt to use radical scavengers to confirm the active species in the reaction. However, using organic radical scavengers might influence the catalytic activity in other ways (e.g. through competitive adsorption). It may have been more useful to use an inorganic scavenger for these experiments. B) Furthermore, the references the authors have used to support the formation of hydroxyl/peroxy radicals in their response document don't seem (to me at least) to be relevant. In Davis and co-workers paper, the authors don't suggest that oxygen reduction by water results in the homolytic cleavage of O₂. This is what what would be require to produce hydroxyl radicals, wouldn't it?

C) The authors state in their revised text that hydroxyl radicals are produced through the interaction of O₂ with H₂... is this correct?

Response:

Thanks for the reviewer's comment.

A) Prompted by the suggestion, we further performed control experiments using the inorganic scavenger of NaCO₃, NaHCO₃ and Ce₂(CO₃)₃. As demonstrated in Supplementary Table 6-2, the introduction of these hydroxyl radical quenchers, significantly decreased glycerol conversion was observed.

B) In response to the suggestion, we added more relevant literature to support the formation of hydroxyl/peroxy radicals [Nature. Comm. 2022, 13, 5467; Phys. Chem. Chem. Phys., 2011, 13, 20178–20187; Science, 2010, 330, 74]. An et al. [Nature. Comm. 2022, 13, 5467] confirmed that O₂ could generate hydroxyl/peroxy radicals in the glycerol oxidation reaction through color rendering experiments. Duan et al. [Phys. Chem. Chem. Phys., 2011, 13, 20178–20187] proved that the O₂ could be reduced to hydroxyl/peroxy radicals via DFT calculation. In Davis and co-workers paper, they stated that “the regeneration of HO⁻ species via O₂ reduction with water, thus closing the catalytic cycle.....Reduction of O₂ by water had

a low barrier of 16 kJ mol^{-1} on Au(111).” Plausibly, our schematic diagram in Fig. 4c has caused a misunderstanding. O_2 does not directly undergo homolytic cleavage. The H obtained from H_2O or other intermediates is necessary to first reduce O_2 and then convert it into OOH^- or OH^- . Based on this, we have revised Fig. 4c and added relevant references to support the conclusion.

C) Thanks for the reviewer’s careful review. It should be the interaction of O_2 with H_2O . We have corrected it in the revised manuscript.

Fig. 4: Oxygen activation into hydroxyl radical on Lewis acid site in MnO₂-D. **a** In situ EPR spectra for the detection of the evolution of oxygen vacancies on MnO₂-D. **b** In situ EPR spectra with free radical trapping agent (DMPO) for the oxidation of glycerol on MnO₂-P and MnO₂-D (for the capitation of hydroxyl radical). **c** Electron transition in orbit during O₂ adsorption over Mn^{δ+}-O_v structure.

Supplementary Table 6-2. Oxidation of glycerol over MnO₂ catalysts

Catalyst	Selectivity (%)					Conversion (%)	Reaction conditions
	GLYA	GLYOA	OA	FA	Others		
(α-)MnO ₂ -P	17.4	30.6	19.5	24.1	8.1	31.6	120°C 6h
(α-)MnO ₂ -T	3.8	8.2	10.7	67.5	8.8	78.7	120°C 6h

(α-) MnO ₂ -D	1.4	3.2	2.8	83.2	8.8	99.2	120°C 6h
(β-) MnO ₂ -H	22.5	11.8	5.8	27.1	30.2	39.1	120°C 6h
(β-) MnO ₂ -L	10.5	5.7	2.5	63.5	15.9	79.0	120°C 6h
(γ-) MnO ₂ -H	15.2	9.1	10.6	26.2	37.1	41.4	120°C 6h
(γ-) MnO ₂ -L	12.5	0.0	5.3	56.5	23.9	80.7	120°C 6h
Only Glycerol	Glycerol conversion: 23.8%					-	120°C 0.5h
Methanol/Glycerol=0.5	Glycerol conversion: 8.9%			Methanol conversion: 34.4%			120°C 0.5h
Isopropanol/Glycerol=0.5	Glycerol conversion: 14.7%			Isopropanol conversion: 54.8%			120°C 0.5h
Tert butanol/Glycerol=0.5	Glycerol conversion: 11.5%			Tert butanol conversion: 20.0%			120°C 0.5h
Na ₂ CO ₃ /Glycerol=0.5	Glycerol conversion: 13.0%					-	120°C 0.5h
Na ₂ HCO ₃ /Glycerol=0.5	Glycerol conversion: 14.5%					-	120°C 0.5h
Ce ₂ (CO ₃) ₃ /Glycerol=0.5	Glycerol conversion: 5.7%					-	120°C 0.5h
MnO ₂ -D(base free)-1	0.5	4.1	2.0	68.9	23.1	29.1	120°C 6h
MnO ₂ -D(base free)-2	0.9	2.3	3.8	61.1	29.5	30.5	120°C 6h

Note: other reaction conditions: 0.1 g catalyst, 25 mL aqueous solution of glycerol (0.1 M), 0.5 g NaOH, 1 MPa O₂.

“The hydroxyl radical quenching experiment, employing organic and inorganic radical scavengers, further established that the produced hydroxyl radical significantly impacts the oxidation reaction (Supplementary Table 6). It is plausible that O₂ mainly interacts with H₂O to generate hydroxyl radicals that participate in the subsequent glycerol activation, aligning with previous studies⁷²⁻⁷⁴.”[Page 12, Line 238-241]

Again, thanks very much for the reviewer’s careful review and valuable suggestions.

Reviewer #2:

I am satisfied with the authors' response. It could be accepted for the revised manuscript.

Response:

We appreciate the positive comments from reviewer.

Reviewer #3:

I am satisfied that the authors have addressed my concerns from the original peer review, particularly regarding the mechanism. The concerns of the other reviewers also appear to have been addressed well. The authors have also extended the study at the request of one of the other reviewers, and overall I think the manuscript is stronger for it. Overall I believe the article is now suitable for acceptance in Nat Comm, and the authors should be commended for doing such a good and thorough job with their revisions.

Response:

We appreciate the positive evaluation from this reviewer.

REVIEWERS' COMMENTS

Reviewer #1 (Remarks to the Author):

I am satisfied with the authors revisions and response to my comments. It's a very nice piece of work and I am happy to recommend it for publication in Nature Communications.